# Accelerating Learned Image Compression Through Modeling Neural Training Dynamics

**Yichi Zhang**                                         *zhan5096@purdue.edu*
*Purdue University*

**Zhihao Duan**                                         *duan90@purdue.edu*
*Purdue University*

**Yuning Huang**                                        *huan1781@purdue.edu*
*Purdue University*

**Fengqing Zhu**                                        *zhu0@purdue.edu*
*Purdue University*

**Reviewed on OpenReview:** *https://openreview.net/forum?id=nannw4SGfS*

## Abstract

As learned image compression (LIC) methods become increasingly computationally demanding, enhancing their training efficiency is crucial. This paper takes a step forward in accelerating the training of LIC methods by modeling the neural training dynamics. We first propose a Sensitivity-aware True and Dummy Embedding Training mechanism (STDET) that clusters LIC model parameters into few separate modes where parameters are expressed as affine transformations of reference parameters within the same mode. By further utilizing the stable intra-mode correlations throughout training and parameter sensitivities, we gradually embed non-reference parameters, reducing the number of trainable parameters. Additionally, we incorporate a Sampling-then-Moving Average (SMA) technique, interpolating sampled weights from stochastic gradient descent (SGD) training to obtain the moving average weights, ensuring smooth temporal behavior and minimizing training state variances. Overall, our method significantly reduces training space dimensions and the number of trainable parameters without sacrificing model performance, thus accelerating model convergence. We also provide a theoretical analysis on the Noisy quadratic model, showing that the proposed method achieves a lower training variance than standard SGD. Our approach offers valuable insights for further developing efficient training methods for LICs.

## 1 Introduction

With the widespread adoption of high-resolution cameras and the increasing prevalence of image-centric social media platforms and digital galleries, images have become a predominant form of media in daily life. The large file sizes of these high-resolution images place considerable demands on both transmission bandwidth and storage capacity. To address these challenges, lossy image compression has emerged as a critical technique for achieving efficient visual communication.

Recently, learned image compression (LIC) methods have garnered significant attention due to their remarkable performance (He et al., 2022; Liu et al., 2023; Li et al., 2024a; Zhang et al., 2024b;a; 2025).

Despite these advancements, LICs currently face a core challenge: high training complexity. Designing a new method requires a substantial amount of computational resources, which severely hinders the emergence of new approaches. Modern LICs typically feature a vast number of parameters that, while enhancing performance, also introduce critical issues, such as an overwhelming computational burden and prolonged

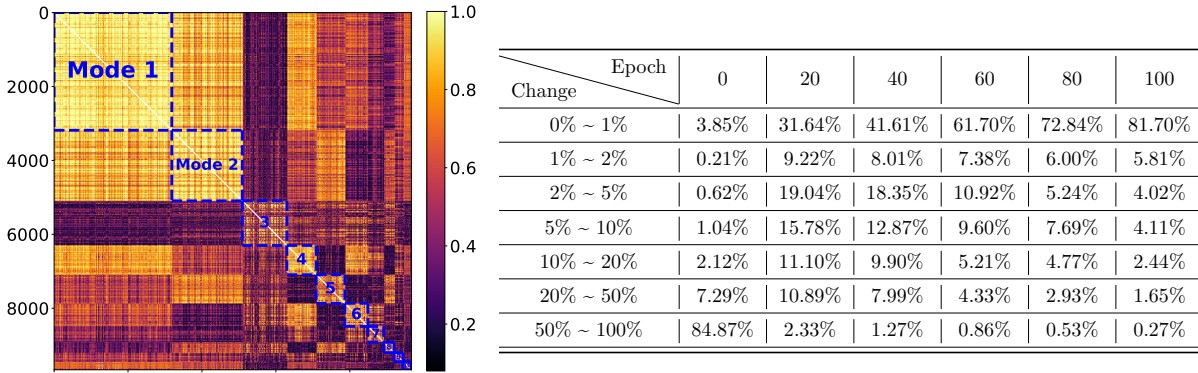

Figure 1: ELIC (He et al., 2022) model, $\lambda = 0.0018$. (a) Clustered correlation matrix of sampled 10k parameter trajectories trained on COCO2017 dataset (Lin et al., 2014), decomposed to 10 modes. The diagonal block structure indicates high correlations of the parameters within each mode, which shows the accurate representation of the proposed method. (b) The table shows the percentage of affine coefficients $\{k_i\}_{i=1}^N$ relative to the total number of coefficients, grouped by relative change intervals, at different epochs during the training of the ELIC model. These relative changes are measured against the final coefficient values at epoch 120. The rows correspond to the relative change intervals (0% - 1%, 1% - 2%, ..., 50% - 100%), indicating how much the coefficients have relatively changed compared to their values at epoch 120. The columns represent specific epochs (0, 20, 40, 60, 80, 100). The percentages indicate the proportion of coefficients that fall within each relative change interval at the corresponding epoch. The table reveals that most coefficients either remain stable or undergo only minor changes as training progresses from epoch 20 to 120. Notably, the proportion of coefficients in the 0% to 1% interval increases significantly from 31.64% at epoch 20 to 81.70% at epoch 100, indicating a marked stabilization of the affine coefficients over time.

training times. For instance, training the FLIC models (Li et al., 2024a) takes up to 520 hours (21 days) using a single 4090 GPU (Tab. 1), while the design process of the FLIC models likely requires even more GPU resources, as it involves extensive experiments, including hyper-parameter tuning, structural adjustments, and other iterative processes. If the training speed of these models is not improved, the time required to develop new methods may become prohibitive. Therefore, developing strategies to reduce training times is of paramount importance.

While there have been efforts to design more efficient LIC methods (e.g., (He et al., 2022; Zhang et al., 2024c)), improving the training efficiency of LIC methods has not received significant attention. A promising and effective direction is to train LICs within low-dimensional subspaces, as has been examined in the context of image classification tasks (Li et al., 2018; Gressmann et al., 2020; Li et al., 2022b;c; Brokman et al., 2024). Instead of exploiting the full parameter space, which can be very large (in millions or even billions), subspace training constrains the training trajectory to a low-dimensional subspace. These approaches are based on the hypothesis that "learned over-parameterized models reside in a low intrinsic dimension" (Li et al., 2018; Aghajanyan et al., 2021). In such subspaces, the degree of freedom for training is substantially reduced (to dozens or hundreds), leading to many favorable properties, such as fast convergence (Li et al., 2023b), robust performance (Li et al., 2022b), and theoretical insights (Li et al., 2018; 2022b).

In this work, we present an innovative approach to improve training efficiency by modeling neural training dynamics in a low dimension space. Our proposed method draws inspiration from Dynamic Mode Decomposition (DMD) (Schmid, 2010; 2022; Brokman et al., 2024; Mudrik et al., 2024), a technique traditionally used in fluid dynamics to decompose complex systems into simpler modes. We extend this concept to LIC by proposing the Sensitivity-aware True and Dummy Embedding Training (STDET) mechanism. The fundamental principle of our approach is the recognition that LIC parameters are highly correlated and can be effectively represented by few distinct "Modes" (see Fig. 1), which capture their intrinsic dimensions. Within each mode, the parameters are aptly modeled through an affine transformation of the reference parameters,

given their significant correlation. We also observe that after the initial head-stage training, the relative changes of the affine coefficients become negligible, and the coefficients tend to remain stable throughout the training, as illustrated in Fig. 1. It is evident that the majority of the coefficients exhibit 0% to 1% relative changes, with the percentage in this interval continuing to increase. This stability permits a focused update of reference parameters and then allows for the "embedding" of non-reference parameters that have relatively invariant coefficients. Once embedded, these parameters are no longer trainable. Subsequently, updates of these embedded parameters are done solely through the fixed coefficients affine transformation of the reference parameters. Non-embedded and reference parameters are still updated via normal training. This mechanism significantly reduces the number of training parameters and the dimension of LIC models in practice along the training period, thereby expediting convergence, as demonstrated in Fig. 2. Ideally, the training parameters and dimensions could be reduced to the number of "Modes". Additionally, to ensure smooth temporal behaviors and minimize training variances, which is the core requirement of STDET, we introduce the well-known moving average techniques to the training phase. Our proposed Sampling-then-Moving Average (SMA) method interpolates the periodically sampled parameter states from SGD training to derive moving averaged parameters, thus enhancing stability and reducing the variance of final model states. Our method is substantiated by comprehensive experiments on various complex LIC models on different domains and comparisons with other efficient training methods. The superiority of the proposed method is also validated through theoretical analysis on the noisy quadratic model.

Our contributions are summarized as follows:

- We propose the Sensitivity-aware True and Dummy Embedding Training (STDET) mechanism, which approximates the SGD training of LICs in a low-dimensional space (Sec. 3.2).

- We introduce Sampling-then-Moving Average (SMA) to ensure the smooth temporal behavior required by STDET, reducing the variances of final states and enhancing training stability (Sec. 3.3).

- We provide a theoretical analysis using the noisy quadratic model to demonstrate the low training variance of the proposed method (Appendix Sec. A.6).

- Overall, our proposed method significantly accelerates the training of LICs while reducing the number of trainable parameters and dimensions without compromising performance (Sec. 4.2, Sec. 4.3, Appendix Sec. A.1).

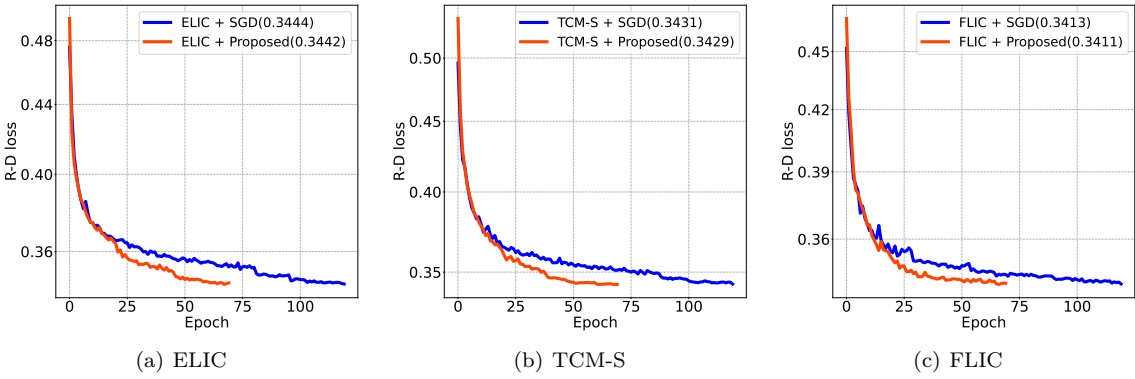

|       | (a) ELIC | (b) TCM-S | (c) FLIC |
|-------|----------|-----------|----------|

Figure 2: **Testing loss comparison of various methods.** *Please zoom in for more details.* The proposed method clearly converges much faster than standard SGD on various LICs. Additionally, as shown in the upper right corner, our method achieves a similar final convergence compared to SGD. $\lambda = 0.0018$, Testing R-D loss $= \lambda \cdot 255^2 \cdot \mathrm{MSE} + \mathrm{Bpp}$. Evaluated on the Kodak dataset.

## 2 Related work

### 2.1 Efficient learned image compression

To address the issue of high computational complexity in learned image compression, various approaches have been proposed: Minnen & Singh (2020) develop a channel-wise autoregressive model to capture channel relationships instead of spatial ones. He et al. (2021) propose a parallelizable two-pass checkerboard model to accelerate spatial autoregressive models. Tao et al. (2023) introduce dynamic transform routing to activate optimally-sized sub-CAEs within a slimmable supernet, conserving computational resources. Duan et al. (2023) incorporate a hierarchical VAE for probabilistic modeling, which generalizes autoregressive models. Guo-Hua et al. (2023) propose an efficient single-model variable-bit-rate codec with mask decay capable of running at 30 FPS with 768x512 input images. Kamisli et al. (2024) implement a variable-bit-rate codec based on multi-objective optimization. Yang & Mandt (2023) adopt shallow or linear decoding transforms to reduce decoding complexity. Ali et al. (2023) introduce a correlation loss that forces the latents to be spatially decorrelated, fitting them to an independent probability model and eliminating the need for autoregressive models. Minnen & Johnston (2023) conduct a rate-distortion-computation study, leading to a family of model architectures that optimize the trade-off between computational requirements and R-D performance. Zhang et al. (2024b) propose contextual clustering to replace computationally intensive self-attention mechanisms. Zhang et al. (2024c) test various transforms and context models to identify a series of efficient LIC models. However, these works primarily focus on test-time efficiency and often overlook the significant computational resources consumed during the training phase. Training complex LICs is resource-intensive, and optimizing training efficiency is crucial.

### 2.2 Low dimension training/fine-tuning of modern neural networks

Training or fine-tuning neural networks in low-dimensional spaces has garnered significant attention recently. Li et al. (2018) propose training neural networks in random subspaces of the parameter space to identify the minimum dimension required for effective solutions. Gressmann et al. (2020) enhance training performance in random bases by considering layerwise structures and redrawing the random bases iteratively. Li et al. (2022b) utilizes top eigenvectors of the covariance matrix to span the training space. Additionally, Li et al. (2022c) apply subspace training to adversarial training, mitigating catastrophic and robust overfitting. Further advancements include Li et al. (2023b), which propose trainable weight averaging to optimize historical solutions in the reduced subspace, and Aghajanyan et al. (2021), which demonstrate that learned over-parameterized models reside in a low intrinsic dimension. Hu et al. (2022) introduce Low-Rank Adaptation based on the premise that weight changes during model adaptation have a low "intrinsic rank". Similarly, Ding et al. (2023) present a delta-tuning method optimizing only a small portion of model parameters, while Jia et al. (2022) propose Visual Prompt Tuning, introducing less than 1% of trainable parameters in the input space. Barbano et al. (2024) constrain deep image prior optimization to a sparse linear subspace of parameters, employing a synergy of dimensionality reduction techniques and second-order optimization methods. These works highlight the elegant performance of the low-dimensional hypothesis, which has not yet been thoroughly explored in the context of learned image compression.

## 3 Proposed method

### 3.1 Preliminaries on learned image compression

Currently, LICs predominantly utilize the non-linear transform coding framework (Ballé et al., 2020), where the transform function and entropy estimation function are parameterized by neural networks. This framework encodes data into a discrete representation for decorrelation and energy compression, and then an entropy model is used to estimate the probability distribution of the discrete representation for entropy coding, as illustrated in Fig. 3. The process begins with an autoencoder applying a learned nonlinear analysis transform, $g_a$, to map the input image $x$ to a lower-dimension latent variable $y = g_a(x)$. This latent variable $y$ is then quantized to $\hat{y}$ using a quantization function $Q$ and encoded into a bitstream via a lossless codec (Duda, 2009), resulting in a much smaller file size. The decoder then reads the bitstream of $\hat{y}$, ap-

plies the synthesis transform $g_s$, and reconstructs the image as $\hat{x}$, with the goal of minimizing the distortion $\mathcal{D}(x, \hat{x})$. Minimizing the entropy of $\hat{y}$ involves learning its probability distribution through entropy modeling, which is achieved using an entropy model $P$ that includes both forward and backward adaptation methods. The forward adaptation employs a hyperprior estimator, which is another autoencoder with its own hyper analysis transform $h_a$ and hyper synthesis transform $h_s$. This hyperprior effectively captures spatial dependencies in the latent representation $y$ and generates a separately encoded latent variable $\hat{z}$, which is sent to the decoder. A factorized density model (Ballé et al., 2018) is used to learn local histograms, estimating the probability mass $p_{\hat{z}}(\hat{z}|\psi_f)$ with model parameters $\psi_f$. The backward adaptation estimates the entropy parameters of the current latent element $\hat{y}_i$ based on previously coded elements $\hat{y}_{<i}$, where $i$ is the latent element index, exploring the redundancy between latent elements. This is achieved through a network $g_b$ that operates in an autoregressive manner over the spatial dimension, channel dimension, or a combination of both. The outputs from $g_b$ and the hyperprior are then used to parameterize the conditional distribution $p_{\hat{y}}(\hat{y}|\hat{z})$ via an entropy parameters network $g_e$. The LIC framework can be formulated as:

$$
\begin{aligned}
\hat{y} &= Q(g_a(x; \phi_a)), \\
\hat{x} &= g_s(\hat{y}; \phi_s), \\
\hat{z} &= Q(h_a(y; \theta_a)), \\
p_{\hat{y}_i}(\hat{y}_i|\hat{z}) &\leftarrow g_e(g_b(\hat{y}_{<i}; \theta_b), h_s(\hat{z}; \theta_s); \psi_e),
\end{aligned}
\tag{1}
$$

with the Lagrange multiplier-based rate-distortion (R-D) loss function $\mathcal{L}$ for end-to-end training:

$$
\begin{aligned}
\mathcal{L}(\phi, \theta, \psi) &= \mathcal{R}(\hat{y}) + \mathcal{R}(\hat{z}) + \lambda \cdot \mathcal{D}(x, \hat{x}), \\
&= \mathbb{E}[\log_2(p_{\hat{y}}(\hat{y}|\hat{z}))] + \mathbb{E}[\log_2(p_{\hat{z}}(\hat{z}|\psi))] + \lambda \cdot \mathcal{D}(x, \hat{x}),
\end{aligned}
\tag{2}
$$

where $\mathcal{D}(x, \hat{x})$ measures the distortion (e.g., MSE) between the original image $x$ and the reconstructed image $\hat{x}$. $\mathcal{R}(\hat{z})$ and $\mathcal{R}(\hat{y})$ represent the rate of $\hat{z}$ and $\hat{y}$, respectively.

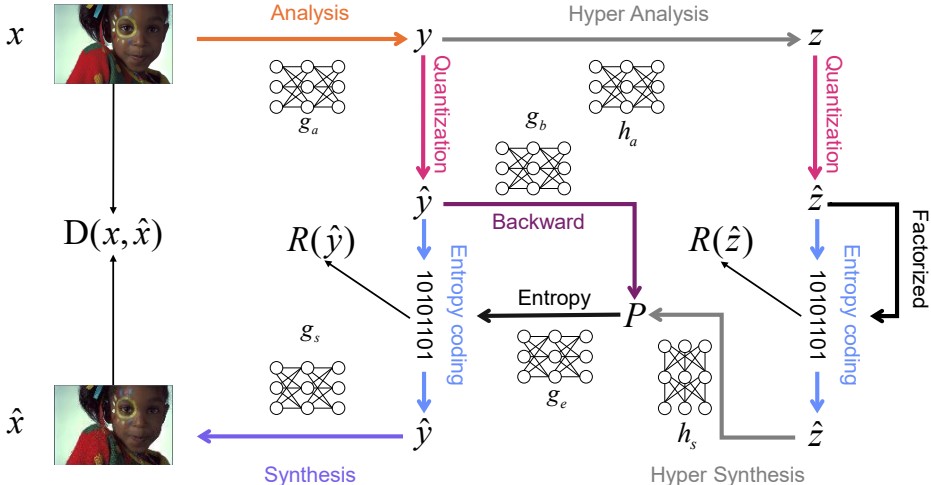

Figure 3: A typical pipeline in learned image compression.

## 3.2 Sensitivity-aware True and Dummy Embedding Training Mechanism

### 3.2.1 Decomposition of non-linear dynamical systems:

As shown in Sec. 3.1 and Fig. 3, LICs heavily rely on neural networks for transform functions and distribution parameters estimation, which are inherently nonlinear and complex systems. However, recent studies (Lusch et al., 2018; Razzhigaev et al., 2024) demonstrate that, under certain conditions, neural networks exhibit intrinsic linear behavior. This linearity can be effectively harnessed using techniques such as Dynamic Mode

Decomposition (DMD) (Schmid, 2010; 2022; Brokman et al., 2024; Mudrik et al., 2024) and Koopman theory (Dogra & Redman, 2020; Brunton et al., 2022), which provide linear approximations of the network's dynamics. By leveraging these methods, the analysis and efficiency of various tasks can be significantly improved. Approximating nonlinear LICs with linear operations could offer numerous benefits, including faster convergence, better interpretability, and greater stability. Therefore, there is substantial potential in employing linear approximations for these complex LICs to achieve enhanced efficiency.

We select a variant of DMD as the core technique for the proposed method due to its simpler and more intuitive representation compared to Koopman theory methods. Let $\mathbf{W} = \{w_i(t)\}_{i=1}^{N-M} \cup \{w_m(t)\}_{m=1}^{M}$ denote the set of LIC's trainable parameters, where $i$ is the index of the non-reference parameters, m is the index of reference parameters, $t$ represents the training time, and $N$ is the total number of parameters. We refer to $\mathbf{W}$ as the set of 'trajectories,' with each $w_i(t)/w_m(t)$ representing an individual 'trajectory.' DMD methods aim to represent the dynamics of complex models by decomposing them into several modes, where each mode is captured by a concise set of reference trajectories $w_m(t)$. This can be expressed as: $w_i(t) \approx \sum_m k_{i,m} w_m(t) + d_i$, where $k_{i,m}$ and $d_i$ are scalars. Specifically, we employ Correlation Mode Decomposition (CMD) (Brokman et al., 2024), which further simplifies this representation:

$$w_i(t) \approx k_i w_m(t) + d_i, \tag{3}$$

where $k_i$ and $d_i$ are scalar affine coefficients associated with $w_i$. CMD operates under the assumption that a complex system can be represented by several modes, and within each mode, an individual parameter can be effectively represented by the affine transformation of the mode reference parameter. This assumption also holds true in the training of LICs, as shown in Fig. 1. CMD can be seen as a special, simplified case of DMDs, where the theoretical dimension corresponds to the number of reference trajectories. With CMD, we achieve a linear approximation of complex nonlinear LICs, which present desirable features.

**Notation:** Let's consider a discrete-time setting in the training process, $t \in [1, \ldots, E]$, where $E$ is the number of epochs. Let $N_m$ be the number of trajectories in mode $m$. Then each parameter $w_m, w_i \in \mathbb{R}^E$, the parameter set $\mathbf{W} \in \mathbb{R}^{N \times E}$, the mode $m$ parameter set $\mathbf{W}_m \in \mathbb{R}^{N_m \times E}$. We use $(t)$ represent at time $t$, and $_t$ represent up to time $t$. We consider 1D vectors such as $u \in \mathbb{R}^E$ to be row vectors, and define the centering operation $\bar{u} := u - \frac{1}{E}\sum_{t=1}^{E} u(t)$, outer product $\langle u, v \rangle = uv^T$, euclidean norm $\|u\| = \sqrt{uu^T}$, and $\mathrm{corr}(u, v) = \frac{\langle \bar{u}, \bar{v} \rangle}{\|\bar{u}\|\|\bar{v}\|}$. Let $K, D \in \mathbb{R}^N$ such that $K(i) = k_i$, $D(i) = d_i$, and let $K_m, D_m \in \mathbb{R}^{N_m}$ be the parts of $K, D$ corresponding to the parameters in mode $m$. Denote matrix form $\tilde{w}_m = \begin{bmatrix} w_m \\ \vec{1} \end{bmatrix} \in \mathbb{R}^{2 \times E}$, where $\vec{1} = (1, 1, \ldots, 1) \in \mathbb{R}^E$, and $\tilde{K}_m = [K_m^T, D_m^T] \in \mathbb{R}^{N_m \times 2}$. In this context, Eq. 3 reads as:

$$\mathbf{W}_m \approx \tilde{K}_m \tilde{w}_m. \tag{4}$$

We then use Eq. 4 and these notations to model the LICs trainable parameters. To clarify the term in the following section, we have two types of parameters: reference and non-reference. Non-reference parameters are further divided into embeddable and non-embeddable.

### 3.2.2 Proposed STDET:

The proposed STDET mechanism builds on the notation and concepts defined above. We perform CMD after an initial head-stage training to identify the reference parameters, extract modes, and select the best hyperparameters. We then estimate the sensitivity of the LIC parameters to determine which parameters are "embeddable". Next, the proposed STDET iteratively updates the affine coefficients based on the updated neural state. Subsequently, we embed the non-reference parameters every few epochs according to the stability of the coefficients and the parameter sensitivity. Our method reduces both the training space dimension and the number of trainable parameters as training moves forward, dynamically optimizing the overall training process.

**Step 1: Mode decomposition.**

We select reference parameters after the initial head-stage training. The LIC is trained normally for a predefined number of epochs $F$ to obtain the trajectories $w_{i,F} \in \mathbf{W}_F$ up to epoch $F$. We then perform

Correlation Mode Decomposition (CMD) (Brokman et al., 2024) to get the reference parameters $w_{m,F}$, the mode of each $w_{i,F} \in \mathbf{W}_F$, and the affine coefficients $\tilde{K}_m(F)$ at epoch $F$ based on Eq. 4. We select the hyperparameters (*i.e.*, the number of modes used) based on the traversed instant CMD performance, as detailed in Sec. 4.4.4 for each model. This CMD determination and association is performed only once. Once $w_{i,F}$ is associated with mode $m$ and reference parameter $w_{m,F}$, $w_i$ remains in mode $m$ with reference parameter $w_m$ for the entire training process.

**Step 2: Parameter Sensitivity Estimation.** Before starting to embed the non-reference parameters in LIC models, it is essential to determine which parameters are "embeddable" as some parameters may be extremely sensitive to perturbations, leading to significant performance drops (Weng et al., 2020; Zhang et al., 2022). We do not embed these parameters. Accurately assessing parameter sensitivity is challenging due to the complexity involved in individually perturbing each parameter and measuring the prediction error. Given the lack of generally reliable estimation methods (Yvinec et al., 2023), we employ a combination of accurate layer-wise assessment and rough parameter-wise estimation to rank the sensitivity of the LIC parameters.

We begin by individually perturbing each layer and measuring the impact on the rate-distortion (R-D) loss using a randomly sampled portion of the training dataset $\mathcal{X} = (x_1, \cdots, x_n)$ (with $n = 256$ in practice). Different strategies are employed for evaluating analysis transform $g_a$, synthesis transform $g_s$ and hyper analysis transform $h_a$, hyper synthesis transform $h_s$, backward network $g_b$, entropy parameters network $g_e$ based on their unique behaviors. For layers in $g_a, g_s$, we compute the R-D loss before and after the perturbation. In contrast, for layers in $h_a, h_s, g_b, g_e$, the perturbation does not affect the reconstruction performance (the $D$ term), as these networks only involve calculating the $R$ term, as shown in Fig. 3. Thus, we only need a single forward pass to obtain $\hat{y}$, and then we individually perturb $h_a, h_s, g_b, g_e$ layers based on the existing $\hat{y}$. This approach allows us to avoid recurrently evaluating $\hat{y}$ and focus solely on changes in $R$, thereby further reducing complexity. For the perturbation, Gaussian noise sampled from $N(0, \sigma^2)$ is added to all parameters when perturbing each layer, where $\sigma$ is a fraction of the maximum parameter value in the layer. After calculating the R-D loss (or $R$ term) increase for all layers, those showing significant increases in R-D loss (or $R$ term) upon perturbation are deemed more sensitive. Following Novak et al. (2018) and Yang et al. (2023), which demonstrate that approximately 75% of parameters can be pruned without significantly affecting performance, we infer that only 25% of the parameters are sensitive. To identify these sensitive parameters, we employ a straightforward and intuitive half-half strategy. We first identify the top 50% most sensitive layers, with 25% from $g_a$ and $g_s$, and 25% from $h_a$, $h_s$, $g_b$, and $g_e$.

Afterward, we estimate the parameter-wise sensitivity for the top 50% sensitive layers' parameters using the first-order estimation (Molchanov et al., 2019):

$$|\mathbf{W}| \odot \nabla : (f, \mathcal{X}) \rightarrow \left( \mathbb{E}_{\mathcal{X}} \left[ |w_i| \cdot \frac{\partial f}{\partial w_i} \right] \right)_{i \in \{1, \ldots, N\}}, \tag{5}$$

where $\odot$ denotes elementwise multiplication. This function combines both the magnitude of the parameter absolute value $|w_i|$ and the gradients of the R-D function $f$ w.r.t. each parameter $\frac{\partial f}{\partial w_i}$, which is effective in practice (Yvinec et al., 2023). By utilizing the estimated rough parameter-wise sensitivity, we identify the 50% relative sensitive parameters from these layers and consider these parameters "non-embeddable" (25% of total parameters), whereas the remaining 75% of parameters are "embeddable". We calculate the sensitivity only once after the head-stage training.

**Step 3. Affine coefficients update.** With the affine coefficients obtained in Step 1 at epoch $F$, we now need to update them for each new epoch to observe their temporal dynamic behaviors. Let $\tilde{K}_m(t)$ be the coefficients $\tilde{K}_m$ evaluated at time $t$. Given the previous epoch $\tilde{K}_m(t-1)$, we can update $\tilde{K}_m$ following Brokman et al. (2024) using Eq. 6:

$$\tilde{K}_m(t) = \left( \tilde{K}_m(t-1)(\tilde{w}_{m,t-1}\tilde{w}_{m,t-1}^T) + W_m(t)\tilde{w}_m^T(t) \right)$$
$$\times (\tilde{w}_{m,t-1}\tilde{w}_{m,t-1}^T + \begin{pmatrix} w_m^2(t) & w_m(t) \\ w_m(t) & 1 \end{pmatrix})^{-1}, \tag{6}$$

where $W_m(t)$ and $\tilde{w}_m(t)$ are the $t$-th columns of $W_m$ and $\tilde{w}_m$, respectively, and $\tilde{w}_{m,t}$ comprises columns 1 through $t$ of $\tilde{w}_m$. Eq. 6 is used for iterative updates after each new epoch.

**Step 4. True and Dummy Embedding.** As we update the coefficients at each new epoch, it is observed that the affine coefficients $k_i$ and $d_i$ tend to stabilize after the head-stage training, as shown in Fig. 1. We can therefore fix $k_i$ and $d_i$ based on their long-term changes $c_i$ evaluated every $L$ epochs. $c_i(t)$ is defined as the Euclidean distance between the current values of $k_i$ and $d_i$ and their $L$ epochs earlier values:

$$c_i(t) = \sqrt{\|k_i(t) - k_i(t-L)\|^2 + \|d_i(t) - d_i(t-L)\|^2}. \tag{7}$$

For $k_i$ and $d_i$ associated with "embeddable" parameters, those with the least $P\%$ changes in terms of $c_i$ are considered embedded coefficients, meaning $k_i$ and $d_i$ are fixed from this point onward. Then, the corresponding $w_i$ are the true embedded non-reference parameters. Note that we do not embed the reference parameters $\{w_m\}_{m=1}^M$. While the embedded parameters are no longer trainable, they still evolve as $w_m(t)$ changes, following $w_i = k_i w_m(t) + d_i$. Consequently, the dimension of the training space and the number of trainable parameters gradually reduce as more and more parameters are embedded. This reduction in the degrees of freedom in the training dynamics facilitates the convergence of the training process. Ultimately, the dimension of the final state training space and the number of trainable parameters converge to the number of modes $M$ in the theoretical case. In practice, we typically embed 50% of the parameters in our experiments.

Additionally, we reinitialize additional $\frac{tP}{2}\%$ parameters that have the least $c_i$ changes, excluding the truly embedded parameters. We assign these parameters the values from their embedding versions: $w_i \leftarrow k_i w_m(t) + d_i$. We refer to this process as dummy embedding. In the next epoch, the dummy embedded parameters are updated like the non-embedded parameters. Empirical results indicate that dummy embedding enhances performance. One explanation is that this dummy embedding scheme simulates the Random Weight Perturbation (RWP) (Li et al., 2024c; Kanashiro Pereira et al., 2021), which has been shown to improve performance and generalization. Throughout the entire training process, we execute Step 1 and Step 2 once after the predefined epoch $F$. Subsequently, we repeatedly perform Step 3 and Step 4 at each new training epoch to embed parameters. The complete detailed STDET algorithm and CMD calculation are detailed in Appendix Sec. A.7 in Algorithm 1.

### 3.3 Sampling-then-Moving Average

#### 3.3.1 Moving average:

As described in Sec. 3.2, the proposed STDET relies heavily on stable correlation and smooth temporal behavior when embedding parameters, necessitating reduced training variance. However, due to the inherent complexity of training LIC models, maintaining such stability can be challenging. To regulate this, we incorporate the moving average technique directly into the training phase, which helps to smooth out noise and randomness to ensure the consistency and stability of the training parameters (Martens, 2020; Chen et al., 2021; Morales-Brotons et al., 2024).

The moving average technique is a straightforward and efficient method for consolidating multiple neural states into a single one, thereby enhancing performance, robustness, and generalization (Li et al., 2023c). It is particularly effective in models that exhibit a degree of similarity across states and does not introduce an unbearable additional computational overhead. Among the most commonly used techniques is the Exponential Moving Average (EMA) (Szegedy et al., 2016; Polyak & Juditsky, 1992).

EMA integrates early-stage states while assigning higher importance to more recent ones, which allows the model to adapt more quickly to changes during training. The EMA parameters are calculated as follows:

$$w_{\text{EMA}}(0) = w(0), \ w_{\text{EMA}}(t+1) = (1-\alpha)w_{\text{EMA}}(t) + \alpha w(t+1), \tag{8}$$

where $\alpha$ is a moving average factor that determines the weight given to recent versus older states.

Moving Average is argued to find flatter solutions in the loss asymmetric valleys than SGD, thus generalizing better to unseen data, with a potential explanation that the loss function near a minimum is often asymmetric, sharp in some directions, and flat in others. While SGD tends to land near a sharp ascent, averaging iterates biased solutions towards a flat region (He et al., 2019).

### 3.3.2 Proposed SMA:

While EMA is typically used during the testing phase, where its parameters do not influence training and only serve as the final model state, some methods (He et al., 2020; Song et al., 2023) integrate EMA into the training process. They achieve this by explicitly computing loss functions that encourage the alignment of SGD trajectories with EMA trajectories, thereby enhancing consistency and stability. In contrast, our approach follows a more straightforward and simpler design. Rather than introducing additional alignment losses, we directly synchronize the EMA-updated sampled SGD parameters (SMA) to the SGD parameters. This enables smoother training dynamics while keeping low complexity.

A simple illustrative comparison of EMA (Szegedy et al., 2016; Polyak & Juditsky, 1992), explicit alignment (Song et al., 2023), and SMA is provided in Fig. 16 in Appendix Sec. A.4.

Specifically, we sample states from the SGD trajectories at regular intervals $l$ and use the moving average rule to update the training parameters. Following EMA principles, the proposed Sampling-then-Moving Average (SMA) maintains a set of SGD parameters $w$ and a set of SMA parameters $w_{\text{SMA}}$. The SGD parameters are obtained by applying the SGD optimization algorithm to batches of training examples. After every $l$ optimizer updates using SGD, we sample the current states and update the SMA parameters through interpolation based on EMA rules as in Eq. 8. Each time the SGD parameters are sampled and the SMA parameters are updated, we synchronize the SMA parameters to the SGD parameters to give a new starting point. The trajectory of the SMA parameters $w_{\text{SMA}}$ is thus characterized as an exponential moving average of the sampled SGD parameters states $w_l$. After $l$ optimizer steps, we have:

$$
\begin{aligned}
w_{\text{SMA}}(t+1) &= (1-\alpha)w_{\text{SMA}}(t) + \alpha w_l(t) \\
&= \alpha\big[w_l(t) + (1-\alpha)w_l(t-1) + \cdots \\
&\quad + (1-\alpha)^{t-1}w_l(0)\big] + (1-\alpha)^t w_{\text{SMA}}(0),
\end{aligned}
\tag{9}
$$

where $\alpha$ is the moving average factor. The SMA parameters leverage recent states from SGD optimization while retaining some influence from earlier SGD parameters to effectively regulate temporal behavior and reduce variance. The additional moving average update and synchronization introduce negligible complexity. We further reduce the complexity by treating the sampling and updating as a single optimizer step, ensuring that the total number of iterations within each epoch remains unchanged. The SMA technique can be implemented as a simplified Lookahead optimizer (Zhang et al., 2019b; Zhou et al., 2021a).

By jointly applying STDET and SMA, the training trajectories of non-reference parameters $w_i$ are modeled as follows:

$$
w_i(t) \leftarrow
\begin{cases}
k_i w_m(t) + d_i & \text{if } w_i(t) \text{ is embedded} \\
\text{SMA update} & \text{else}
\end{cases}
\tag{10}
$$

The reference parameters are updated using SMA.

## 4 Experimental results

### 4.1 Experimental settings

**Training.** We use the COCO2017 dataset (Lin et al., 2014) for training, which contains 118,287 images, each having around $640{\times}420$ pixels. We randomly crop $256{\times}256$ patches from these images.

All models are trained using the Lagrange multiplier-based rate-distortion loss as defined in Eq. 2.

Following the settings of CompressAI (Bégaint et al., 2020) and standard practice (He et al., 2022; Liu et al., 2023; Li et al., 2024a), we set $\lambda$ to $\{18, 35, 67, 130, 250, 483\} \times 10^{-4}$. For all models in the "+ SGD" series, we train each model using the Adam optimizer[1] with $\beta_1 = 0.9$ and $\beta_2 = 0.999$. The $\lambda = 0.0018$ models are trained for 120 epochs. For models with other $\lambda$ values, we fine-tune the model trained with $\lambda = 0.0018$ for an additional 80 epochs. For all models in the "+ Proposed" series, we train each $\lambda = 0.0018$ model using

---

[1]Adam is essentially SGD with first-order moments (*i.e.*, mean) and second-order moments (*i.e.*, variance) estimation.

the Adam optimizer for 70 epochs. For models with other $\lambda$ values, we fine-tune the model trained with $\lambda = 0.0018$ for an additional 50 epochs. More details are provided in the Appendix Sec. A.8.

**Testing.** Three widely-used benchmark datasets, including Kodak[2], Tecnick[3], and CLIC 2022[4], are used to evaluate the performance of the proposed method. We further demonstrate the robustness of the proposed method and its capacity for real-world application by conducting experiments on stereo images, remote sensing images, screen content images, and raw image compression, as shown in Appendix Sec. A.1.

### 4.2 Quantitative results

We compare our proposed method with standard SGD-trained models on prevalent complex LICs, including ELIC (He et al., 2022), TCM-S (Liu et al., 2023), and FLIC (Li et al., 2024a) to demonstrate its superior performance. We use SGD-trained models as the anchor to calculate BD-Rate (Bjøntegaard, 2001).

Tab. 1 presents the BD-Rate reduction of the proposed method compared to the SGD anchors across three datasets. Our proposed method consistently achieves comparable final results to all methods across these datasets, which include both normal and high-resolution images, demonstrating its effectiveness. For example, on the Kodak dataset, "ELIC + Proposed" achieves -0.71% against "ELIC + SGD". Fig. 4 further illustrates the R-D curves of all methods, showing that the R-D points of the SGD-trained model and the model trained by the proposed method almost overlap.

Table 1: Computational Complexity and BD-Rate Compared to SGD

| Method | | Training time ↓ | Total trainable params ↓ | Final trainable params ↓ | BD-Rate (%) ↓ | | |
| --- | --- | --- | --- | --- | --- | --- | --- |
| | | | | | Kodak | Tecnick | CLIC2022 |
| ELIC (He et al., 2022) | + SGD | 165h | 18,418M | 35.42M | 0% | 0% | 0% |
| | + Proposed | **101h** | **9,519M** | **20.66M** | **-0.71%** | **-0.70%** | **-0.61%** |
| TCM-S (Liu et al., 2023) | + SGD | 346h | 23,493M | 45.18M | 0% | 0% | 0% |
| | + Proposed | **213h** | **12,142M** | **26.34M** | **-0.77%** | **-0.74%** | **-0.64%** |
| FLIC (Li et al., 2024a) | + SGD | 520h | 36,899M | 70.96M | 0% | 0% | 0% |
| | + Proposed | **320h** | **19,070M** | **41.39M** | **-0.65%** | **-0.51%** | **-0.63%** |

**Training Conditions**:$1 \times$ Nvidia 4090 GPU, i9-14900K CPU, 128GB RAM. **Bold** represents better performance.

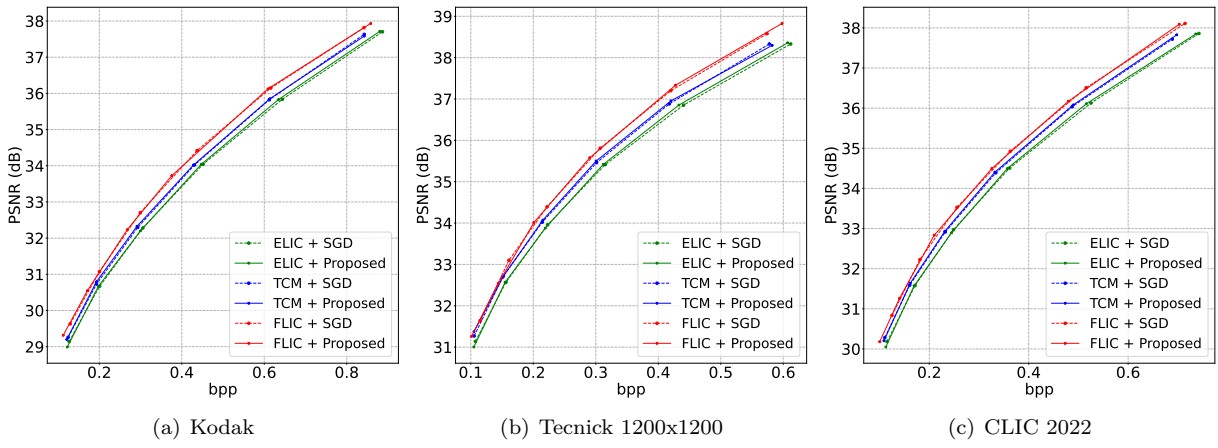

(a) Kodak  (b) Tecnick 1200x1200  (c) CLIC 2022

Figure 4: **R-D curves of various methods.** *Please zoom in for more details.*

We also compare our approach with other efficient low-dimensional training methods, including P-SGD (Li et al., 2022b), P-BFGS (Li et al., 2022b), TWA (Li et al., 2023b), and sparsity-based training methods, such as RigL (Evci et al., 2020) and SRigL (Lasby et al., 2024). P-SGD employs SGD in a projected subspace

---

[2]https://r0k.us/graphics/kodak/

[3]https://tecnick.com/?aiocp%20dp=testimages

[4]http://compression.cc/

derived from PCA. P-BFGS uses the quasi-Newton method BFGS within the same projected subspace. TWA utilizes Schmidt orthogonalization to build the subspace based on historical solutions. RigL (Evci et al., 2020) updates sparse network topology via parameter magnitudes and occasional gradient computations, and SRigL (Lasby et al., 2024) is an improved variant emphasizing dynamic sparsity management.

For these compared low-dimensional methods, we follow the original paper and set the SGD pretrained epoch as 50, after which we train for another 20 epochs using these methods. For the sparsity-based training methods, we train all the models for 70 epochs with a target sparsity of 50% following the original implementations. In total, the training epochs for all the compared methods and our proposed method are set to 70 to ensure a fair comparison.

As shown in Tab. 2, although these low-dimensional methods operate within significantly reduced training dimensions, they do not perform well in terms of performance. Both P-SGD and TWA result in loss divergence, ultimately causing the training to crash. P-BFGS also struggles to converge, leading to a substantially higher R-D loss compared to our proposed method, 0.3982 vs 0.3442. Several factors contribute to these outcomes: (1) These methods were originally designed for image classification tasks and typically use small, simple datasets such as CIFAR10 and CIFAR100, whereas LIC training is inherently more complex; (2) The dimensionality reduction in these methods heavily depends on historical solutions (sampled epochs), limiting the dimensionality to around 50. Directly projecting extremely complex networks into such small dimensions appears to cause non-convergence. Additionally, we observe that P-BFGS does not save training time compared to other efficient training methods, as the quasi-Newton techniques involve time-consuming estimation and iterative updates of the Hessian matrix.

Sparsity-based training methods, despite their efficiency in reducing inference FLOPs, do not accelerate the convergence of LICs. This is primarily due to their growth-and-drop mechanism, which maintains a constant sparsity ratio without reducing the training space dimensions. As a result, they fail to improve convergence speed. Both RigL and SRigL exhibit significantly higher losses compared to SGD and our proposed method. For instance, SRigL's R-D loss is 0.3601, while RigL's loss is 0.3624. Although sparsity-based methods achieve a notable 50% reduction in inference FLOPs compared to all low-dimensional methods and SGD, their poor R-D performance renders this advantage ineffective. This outcome aligns with expectations, as these methods were originally designed for training sparse models in image classification tasks—a fundamentally different objective from our goal of accelerating LIC training.

The comparison with both low-dimensional training methods and sparsity-based approaches clearly highlights the superiority of our proposed method in accelerating LIC convergence without compromising performance.

Table 2: Comparison with Various Efficient Training Methods

| Settings | Training epoch ↓ | Training space dimension ↓ | Total training time ↓ | Inference FLOPs ↓ | R-D loss ↓ |
|---|---|---|---|---|---|
| ELIC + SGD | 120 | 35,424,505 | 38h | 437.6G | 0.3444 |
| ELIC + P-SGD (Li et al., 2022b) | **70** | **40** | **23h** | 437.6G | div. |
| ELIC + P-BFGS (Li et al., 2022b) | **70** | **40** | 35h | 437.6G | 0.3982 |
| ELIC + TWA (Li et al., 2023b) | **70** | 50 | **23h** | 437.6G | div. |
| ELIC + RigL (Evci et al., 2020) | **70** | 35,424,505 | **23h** | 218.8G | 0.3624 |
| ELIC + SRigL (Lasby et al., 2024) | **70** | 35,424,505 | 24h | 214.4G | 0.3601 |
| ELIC + Proposed | **70** | 50* | **23h** | 437.6G | **0.3442** |

**Train Conditions:** $1 \times$ Nvidia 4090 GPU, i9-14900K CPU, 128GB RAM. "div." indicates that these methods result in loss divergence, eventually causing the training to crash. *: The training dimension of our proposed method continues to decrease as training proceeds, theoretically converging to 50. $\lambda = 0.0018$. R-D loss $= \lambda \cdot 255^2 \cdot \text{MSE} + \text{Bpp}$. **Bold** indicates the best.
FLOPs are calculated using the ptflops library with an input image size of $512 \times 512$ pixels.

## 4.3 Complexity

We evaluate the complexity of SGD and our proposed method by measuring total training time, total trainable parameters for the whole training process, and trainable parameters in the final epoch, as presented in Tab. 1. Total training time is defined as the time required to train all models for each $\lambda$ value on a single 4090 GPU. Compared to models trained using SGD, our proposed method is significantly more efficient,

reducing the training time to approximately 62% of that required by SGD. This efficiency is mainly attributed to the faster convergence of our method, which achieves the desired performance with fewer epochs while maintaining comparable training time per epoch. For example, training all FLIC models using SGD takes 520 hours ( 21 days on a single 4090 GPU), which is unbearable. In contrast, our proposed method drastically reduces the training time to 320 hours (13 days). Similarly, the training time for TCM-S is reduced from 346 hours (14 days) to 213 hours (8 days), highlighting the clear advantages of our approach.

The total trainable parameters are calculated by multiplying the model's trainable parameters by the total number of training epochs across all $\lambda$ values. For example, in the case of "ELIC + SGD", the total trainable parameters are computed as $35.42\text{M} \times 520$ epochs, resulting in $18,418\text{M}$. For "ELIC + Proposed", using $\lambda = 0.0018$ as an example, the total trainable parameters from epoch 1 to 20 are $35.42\text{M} \times 20$ epochs. Starting from epoch 21, we begin embedding 1% of the parameters in each epoch. Thus, for epochs 21 to 70, the total trainable parameters are calculated as $100\% \times 35.42\text{M} + 99\% \times 35.42\text{M} + \cdots + 51\% \times 35.42\text{M}$. By performing similar calculations for all other $\lambda$ values, we find that the total trainable parameters for "ELIC + Proposed" are $9,519\text{M}$, approximately 51% of the total trainable parameters required by the SGD method. Similarly, the total trainable parameters for FLIC and TCM-S methods are reduced from $36,899\text{M}$ (FLIC) and $23,493\text{M}$ (TCM-S) to $19,070\text{M}$ (FLIC) and $12,142\text{M}$ (TCM-S), respectively, demonstrating a substantial reduction. We also report the final trainable parameters, representing the number of trainable parameters after the final epoch. For instance, FLIC's trainable parameters are reduced from 70.96M to 41.39M. By reducing the number of trainable parameters, we possess the potential to further accelerate training by utilizing sparse training methods (Zhou et al., 2021b), which will be discussed in Appendix Sec. A.5.

### 4.4 Ablation study

Hyperparameters are critical to the proposed method. To evaluate the influence of various factors and demonstrate the process of selecting the best hyperparameters, we conducted comprehensive experiments across all three models. In the R-D plane (bpp-PSNR figures), better performance is indicated by points in the upper left region. Figures 5, 6, and 7 depict the R-D curves for ELIC, TCM, and FLIC under different configurations of mode decomposition, embedding mechanisms, and SMA, respectively, using $\lambda = 0.0018$, 0.0035, and 0.0067 as illustrative examples. The corresponding BD-Rate results are summarized in Tables 3, 4, and 5, where the final configuration in each column is used as the anchor for BD-Rate calculations.

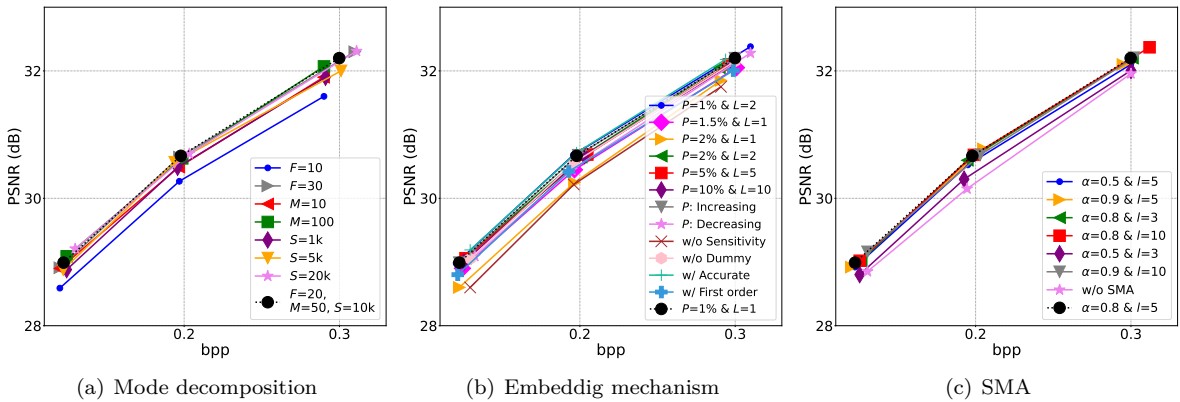

Figure 5: **Ablation experiments on proposed methods.** ELIC model.

### 4.4.1 What is the impact of the factors in the CMD calculation procedure?

In our method, three key factors influence the Step 1 CMD calculation. The first key factor is the predefined epoch $F$, which determines when CMD is performed and the embedding mechanism begins. As shown in Fig. 2, the most significant loss changes occur within the first 20 epochs. To ensure embedding neither starts too early nor too late, we set $F = 20$. Starting the embedding process too early could compromise

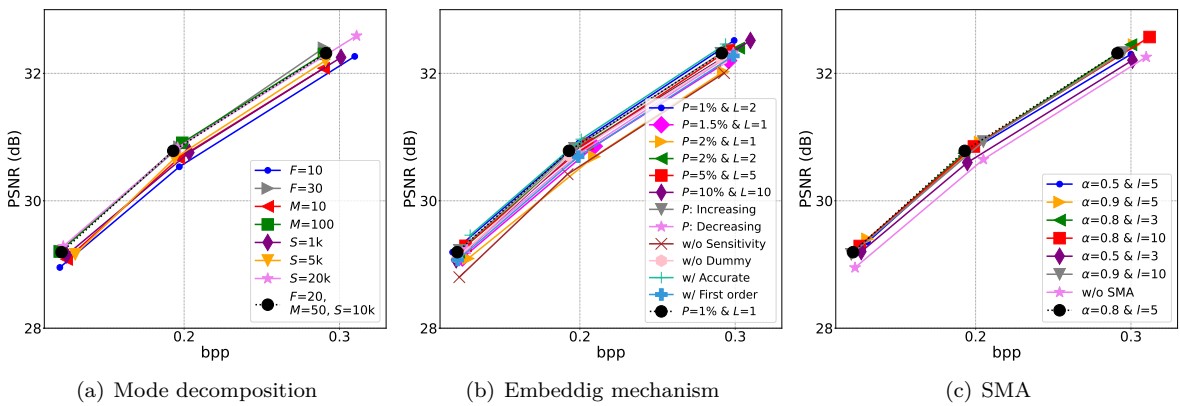

Figure 6: **Ablation experiments on proposed methods.** TCM-S model.

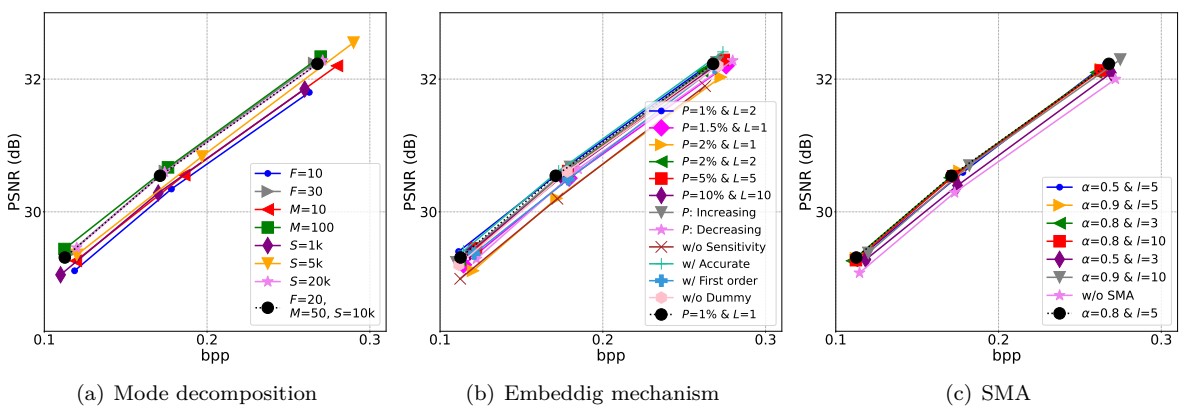

Figure 7: **Ablation experiments on proposed methods.** FLIC model.

Table 3: Ablation study of hyperparameters on ELIC model. BD-Rate values show coding efficiency compared to the bottom row baseline configurations (marked with 0%). Lower BD-Rate denotes better.

| Mode Decomposition | | Embedding | | SMA | |
|---|---|---|---|---|---|
| **Setting** | **BD-Rate** | **Setting** | **BD-Rate** | **Setting** | **BD-Rate** |
| $F = 10$ | 11.54% | $P = 1\%$ & $L = 2$ | -1.40% | $\alpha = 0.5$ & $l = 5$ | 2.48% |
| $F = 30$ | 0.24% | $P = 1.5\%$ & $L = 1$ | 5.20% | $\alpha = 0.9$ & $l = 5$ | 0.48% |
| $M = 10$ | 3.83% | $P = 2\%$ & $L = 1$ | 10.72% | $\alpha = 0.8$ & $l = 3$ | 0.66% |
| $M = 100$ | 0.63% | $P = 2\%$ & $L = 2$ | 1.43% | $\alpha = 0.8$ & $l = 10$ | 0.26% |
| $S = 1$k | 4.34% | $P = 5\%$ & $L = 5$ | 2.34% | $\alpha = 0.5$ & $l = 3$ | 7.39% |
| $S = 5$k | 1.85% | $P = 10\%$ & $L = 10$ | 2.24% | $\alpha = 0.9$ & $l = 10$ | 1.75% |
| $S = 20$k | 0.66% | $P$: Increasing | 0.15% | $\alpha = $ w/o SMA | 11.91% |
| $F = 20, M = 50, S = 10$k | 0% | w/o Sensitivity | 12.34% | $\alpha = 0.8$ & $l = 5$ | 0% |
| - | - | w/o Dummy | 2.29% | - | - |
| - | - | w/ Accurate | -1.21% | - | - |
| - | - | w/ First order | 4.55% | - | - |
| - | - | $P = 1\%$ & $L = 1$ | 0% | - | - |

Table 4: Ablation study of hyperparameters on TCM-S model. BD-Rate values show coding efficiency compared to the bottom row baseline configurations (marked with 0%). Lower BD-Rate denotes better.

| Mode Decomposition | | Embedding | | SMA | |
|---|---|---|---|---|---|
| **Setting** | **BD-Rate** | **Setting** | **BD-Rate** | **Setting** | **BD-Rate** |
| $F = 10$ | 8.88% | $P = 1\%$ & $L = 2$ | -0.83% | $\alpha = 0.5$ & $l = 5$ | 2.15% |
| $F = 30$ | -0.40% | $P = 1.5\%$ & $L = 1$ | 5.93% | $\alpha = 0.9$ & $l = 5$ | 1.03% |
| $M = 10$ | 5.46% | $P = 2\%$ & $L = 1$ | 10.42% | $\alpha = 0.8$ & $l = 3$ | 0.62% |
| $M = 100$ | -0.65% | $P = 2\%$ & $L = 2$ | 1.62% | $\alpha = 0.8$ & $l = 10$ | 0.92% |
| $S = 1$k | 5.62% | $P = 5\%$ & $L = 5$ | 2.22% | $\alpha = 0.5$ & $l = 3$ | 5.94% |
| $S = 5$k | 4.35% | $P = 10\%$ & $L = 10$ | 2.29% | $\alpha = 0.9$ & $l = 10$ | 1.46% |
| $S = 20$k | -0.15% | $P$: Increasing | 0.45% | $\alpha = $ w/o SMA | 9.72% |
| $F = 20$, $M = 50$, $S = 10$k | 0% | w/o Sensitivity | 10.42% | $\alpha = 0.8$ & $l = 5$ | 0% |
| - | - | w/o Dummy | 2.40% | - | - |
| - | - | w/ Accurate | -1.38% | - | - |
| - | - | w/ First order | 5.16% | - | - |
| - | - | $P = 1\%$ & $L = 1$ | 0% | - | - |

Table 5: Ablation study of hyperparameters on FLIC model. BD-Rate values show coding efficiency compared to the bottom row baseline configurations (marked with 0%). Lower BD-Rate denotes better.

| Mode Decomposition | | Embedding | | SMA | |
|---|---|---|---|---|---|
| **Setting** | **BD-Rate** | **Setting** | **BD-Rate** | **Setting** | **BD-Rate** |
| $F = 10$ | 10.53% | $P = 1\%$ & $L = 2$ | -0.92% | $\alpha = 0.5$ & $l = 5$ | 1.78% |
| $F = 30$ | -0.48% | $P = 1.5\%$ & $L = 1$ | 5.48% | $\alpha = 0.9$ & $l = 5$ | 0.45% |
| $M = 10$ | 7.67% | $P = 2\%$ & $L = 1$ | 10.36% | $\alpha = 0.8$ & $l = 3$ | 0.13% |
| $M = 100$ | -1.55% | $P = 2\%$ & $L = 2$ | 1.30% | $\alpha = 0.8$ & $l = 10$ | 0.83% |
| $S = 1$k | 7.43% | $P = 5\%$ & $L = 5$ | 2.06% | $\alpha = 0.5$ & $l = 3$ | 6.17% |
| $S = 5$k | 4.86% | $P = 10\%$ & $L = 10$ | 2.31% | $\alpha = 0.9$ & $l = 10$ | 1.77% |
| $S = 20$k | 0.01% | $P$: Increasing | 0.73% | $\alpha = $ w/o SMA | 8.96% |
| $F = 20$, $M = 50$, $S = 10$k | 0% | w/o Sensitivity | 10.73% | $\alpha = 0.8$ & $l = 5$ | 0% |
| - | - | w/o Dummy | 2.15% | - | - |
| - | - | w/ Accurate | -1.35% | - | - |
| - | - | w/ First order | 5.11% | - | - |
| - | - | $P = 1\%$ & $L = 1$ | 0% | - | - |

the accuracy of the computed reference parameters and sensitivity estimations due to ongoing significant changes, negatively impacting performance. Conversely, starting too late may delay convergence acceleration and reduce the number of embedded parameters, as fewer epochs remain for training. To evaluate the effects of different $F$ values, we tested $F = 10$, $F = 20$, and $F = 30$, with results reported in Fig. 5(a), 6(a), 7(a), the first column of Tab. 3, 4, and 5. The findings indicate that $F = 10$ leads to significantly worse convergence, e.g., with an 8.88% BD-Rate increase compared to the default $F = 20$ for the TCM-S model. Meanwhile, $F = 30$ achieves only a marginal 0.40% BD-Rate reduction but requires more trainable parameters. Based on these results, we conclude that setting $F = 20$ provides the balance between accuracy and efficiency.

The second critical factor is the number of modes $M$, which plays a vital role in CMD computations. Too few modes fail to capture the correlations between parameters accurately, adversely affecting performance. On the other hand, too many modes overcomplicate the decomposition process and increase memory requirements, particularly during the calculation of correlations between parameters and reference parameters. We conducted experiments with $M = 10$, $M = 50$, and $M = 100$, as shown in the figures and tables. The results suggest that the optimal $M$ value may vary depending on the model. For instance, $M = 50$ delivers the best results for the ELIC model, while $M = 100$ performs best for the TCM-S and FLIC models. Further analysis of this hyperparameter is necessary.

Finally, to address the computational challenge of calculating the large $N \times N$ correlation matrix for identifying reference parameters, we uniformly sample $S$ trajectories to determine the reference trajectories. We compare results using $S = 1k$, $S = 5k$, $S = 10k$, and $S = 20k$, as shown in the figures and tables. Similar to $M$, the results do not clearly indicate the optimal number of sampled trajectories. For example, $S = 10k$ yields the best results for the ELIC model, whereas $S = 20k$ is the best for the TCM-S model. $S = 10k$ and $S = 20k$ shows negligible difference for FLIC model. Therefore, in CMD calculations, the predefined epoch $F$ can be set to 20 across all models, while the number of modes $M$ and the number of sampled trajectories $S$ should be model-specific. Further guidance on efficiently determining these parameters is provided in Sec. 4.4.4.

### 4.4.2   What is the impact of the factors of the embedding mechanism?

Determining the parameter sensitivity is a crucial process in the embedding mechanism. As mentioned earlier, embedding certain parameters may significantly reduce the performance. Therefore, we first add experiments that omit sensitivity estimation, and we observe a substantial decrease in the R-D performance, see Fig. 5(b), 6(b), 7(b), the second column of Tab. 3, 4, and 5 "w/o Sensitivity". This significant change supports our hypothesis that certain parameters are extremely sensitive and inappropriate embedding leads to degraded performance.

To further demonstrate the effectiveness of our proposed combined strategy—half accurate layer-wise assessment and half rough parameter-wise estimation—we include two additional variants. The first variant, pure accurate layer-wise assessment ("w/ Accurate"), perturbs each LIC parameter individually, requiring millions of iterations to estimate sensitivity precisely. While this provides an upper bound for our combined strategy and offers an additional 1.2% BD-Rate improvement, it is computationally infeasible due to its high complexity. The second variant, first-order approximation ("w/ First order"), estimates parameter sensitivity using only a rough first-order approximation. However, as shown in the tables, this method leads to inaccurate sensitivity estimation and consequently suboptimal performance, with an average BD-Rate increase of around 5.5% across all three methods.

Our combined strategy strikes a balance between efficiency and effectiveness, achieving superior performance while avoiding the computational burden of exhaustive sensitivity analysis. These results highlight the advantages of our proposed strategy.

Other factors influencing the embedding mechanism include the embedding percentage, $P$, and the embedding period, $L$. The percentage $P$ dictates the proportion of parameters to be embedded, while $L$ determines the frequency of embedding over specified epochs. As demonstrated in figures and tables, experiments reveal that embedding 1% per epoch ($P = 1\%$ & $L = 1$) and 1% every two epochs ($P = 1\%$ & $L = 2$) yield the best results. $P = 1\%$ & $L = 2$, however, results in too few parameters being embedded. Embedding 1.5%/2% in a single epoch ($P = 1.5\%$ & $L = 1 / P = 2\%$ & $L = 1$) leads to poorer performance due to excessive embedding

in the training process, resulting in inaccurate modeling and a decreased R-D performance. Comparing short-term small embedding percentages ($P = 1\%$ & $L = 1$ and $P = 2\%$ & $L = 2$) with long-term large ones ($P = 5\%$ & $L = 5$ and $P = 10\%$ & $L = 10$), we find that a shorter, smaller setup performs better. This likely stems from the disruptive effect of embedding a large number of parameters simultaneously, which adversely impacts training. Further experiments involving either a linear increase or a decrease schedule for $P$ (We set $P$ appropriately so that the embedded parameters in the final epoch account for 50% of the model's trainable parameters.) show that gradually increasing the embedding percentage yields a performance similar to $P = 1\%$, $L = 1$, but results in more total trainable parameters. Conversely, a gradual decrease leads to poor performance. Therefore, we choose the straightforward approach of $P = 1\%$, $L = 1$ due to its simplicity and effectiveness.

The proposed dummy embedding involves reinitializing the original parameters using the embedded version during the embedding process, akin to the RWP technique, which has proven effective in enhancing performance. To evaluate the efficacy of dummy embedding, we conducted experiments in its absence. The results, detailed as "w/o Dummy", indicate an around 2% BD-Rate performance drop upon removal of dummy embedding, thus affirming its positive impact on performance.

### 4.4.3   What is the impact of the factors of the SMA?

We also conduct extensive experiments to assess the two critically important factors of SMA, as detailed in Fig. 5(c), 6(c), 7(c), the last column of Tab. 3, 4, and  5. A setting of $\alpha = 0.5, l = 5$ results in poorer performance, possibly due to an over-reliance on past states. In contrast, the setting of $\alpha = 0.9, l = 5$ shows no significant difference compared with the utilized $\alpha = 0.8, l = 5$. When we adjust $l$, the values 3, 5, and 10 yield consistent performance. In our next experiments with new combinations of $\alpha$ and $l$, we find that $\alpha = 0.5, l = 3$ performs the worst. $\alpha = 0.9, l = 10$ is slightly less effective than $\alpha = 0.8, l = 5$. We then choose the setting $\alpha = 0.8$, $l = 5$ based on these results. Finally, we also attempted to remove the SMA, "w/o SMA". As previously discussed, the SMA is crucial for ensuring that the STDET operates smoothly. It is evident that the absence of the SMA results in a noticeable performance decline. We also individually examine the roles of the proposed STDET and SMA in Sec. A.3.

From the comprehensive ablation study, we observe that only the number of modes $M$ and the number of sampled trajectories $S$ yield inconsistent results. For all other parameters, a unified configuration can be applied regardless of the model or $\lambda$. The final unified settings are: predefined epoch $F = 20$, embedding percentage $P = 1\%$, embedding period $L = 1$, moving average factor $\alpha = 0.8$, and optimizer step $l = 5$.

### 4.4.4   How to select the number of modes and the number of sampled parameters?

Based on the preliminary results in Sec. 4.4.1, our initial hypothesis suggests that larger models generally require a greater number of modes and sampled trajectories. To validate this hypothesis, we conducted a comprehensive set of experiments across all three models, evaluating $M \in \{10, 30, 50, 70, 100, 120\}$ (number of modes) and $S \in \{2k, 6k, 10k, 14k, 20k, 24k\}$ (number of sampled trajectories). The results are summarized in Fig. 8. For consistency, the anchor setting for each model is $M = 50$ and $S = 10k$.

From the results, we can draw the following conclusions:

1. **Larger models require larger $M$ and $S$, but the performance saturates beyond certain thresholds.**
   For instance:

   - **ELIC** (35.42M parameters): Performance caps at $M = 50$ and $S = 10k$.
   - **TCM-S** (45.18M parameters): Performance reaches its max value at $M = 70$ and $S = 14k$.
   - **FLIC** (70.96M parameters): Performance saturates at $M = 100$ and $S = 20k$.

   Beyond these thresholds, further increases in $M$ and $S$ provide negligible performance gains.

2. **The highest performance for each number of modes is achieved when the number of sampled trajectories is roughly 200 times the number of modes.**
   This corresponds to the diagonal in the figure. For example:

- For $M = 10, 30, 50, 70, 100, 120$, the highest performance is achieved at $S = 2k, 6k, 10k, 14k, 20k, 24k$, respectively for each model.

3. **The number of modes has a greater impact on performance than the number of sampled trajectories.**
   For example:

   - In the FLIC model, with only 10 modes, the performance is as low as 7% BD-Rate.
   - However, even with an inappropriate $S = 2k$, 120 modes can still achieve a BD-Rate of $-0.21\%$.

   Additionally, the lower right region of the figure exhibits deeper colors compared to the upper left region, emphasizing the significant impact of the number of modes.

These findings confirm our initial hypothesis that larger models require a greater number of modes and sampled trajectories, and we only need to consider the diagonal combination. However, in real-world scenarios, it is often impractical to exhaustively search for the best hyperparameters and find their performance cap, even for few combinations. This raises a critical question: How can we determine the optimal number of modes and sampled trajectories without extensive training?

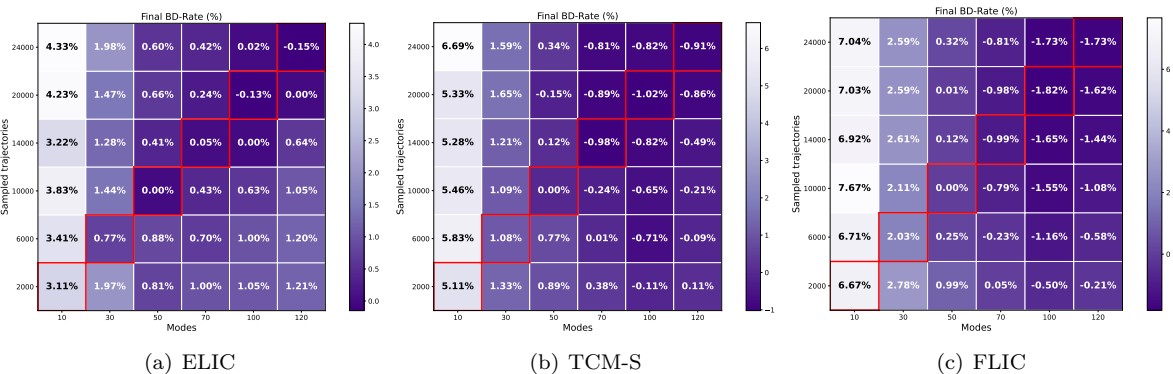

Figure 8: **Ablation experiments on modes $M$ and sampled trajectories $S$.** Deeper color represents better performance.

During our experiment, we observed an intriguing phenomenon: the *CMD instant performance—i.e.*, the performance of the CMD-decomposed model (Step 1) evaluated immediately on a randomly sampled dataset (the same set used in Step 2) after decomposition under certain $M, S$ without additional training—correlated strongly with the final performance achieved after complete training. This observation led us to hypothesize that the quality of CMD decomposition plays a critical role in determining the ultimate model performance, analogous to how initialization methods in neural networks influence optimization and final accuracy (Narkhede et al., 2022; Glorot & Bengio, 2010; He et al., 2015).

To validate this hypothesis, we conducted a systematic study measuring CMD instant performance across various settings of $M$ (number of modes) and $S$ (number of sampled trajectories). The results, shown in Fig. 9, focus on diagonal combinations where $S = 200 \times M$, as suggested by earlier findings. These experiments confirmed that the CMD instant performance improves with increasing $M$ and $S$, eventually stabilizing at specific values. Notably, the saturation points of CMD instant performance closely align with the thresholds where the final performance stabilizes. For example, in the ELIC model, both CMD instant performance and final performance stabilize at $M = 50$ and $S = 10k$. Similar trends are observed for the TCM-S and FLIC models.

This strong correlation suggests that CMD instant performance can serve as a reliable indicator for selecting the optimal values of $M$ and $S$.

Building on this insight, we propose an efficient method for hyperparameter selection:

1. During the CMD decomposition stage, evaluate CMD instant performance for a set of diagonal combinations $(M, S)$ where $S = 200 \times M$. This evaluation requires only a single forward pass per combination, making it computationally lightweight.

2. Identify the combination where CMD instant performance stabilizes or reaches its peak. We define the performance cap as the point where subsequent combinations result in less than a 0.1% R-D loss improvement (approximately corresponding to 0.2% BD-Rate difference). We then use the determined values of $M$ and $S$ for subsequent training.

The detailed procedure is presented in Algorithm 1. This method eliminates the need for exhaustive training to search for optimal hyperparameters, significantly reducing computational costs, as it requires only a forward pass for each combination. Moreover, it is naturally extendable to new models. For instance, for a model with more parameters (e.g., 100M parameters compared to FLIC's 70.96M), diagonal combinations with larger $M$ and $S$ (e.g., $M = 70, 100, 120, 150$ and $S = 200 \times M$) can be explored. Given that FLIC reaches its performance cap at $M = 100$ and $S = 20$k, it is reasonable to expect that larger models will stabilize at proportionally higher values of $M$ and $S$.

This phenomenon parallels the influence of network initialization on final performance in neural networks. Initialization methods such as Xavier (Glorot & Bengio, 2010), Kaiming (He et al., 2015), and orthogonal initialization (Saxe et al., 2013) have demonstrated that the quality of initialization significantly impacts optimization and final accuracy. Similarly, the quality of CMD decomposition serves as a strong initialization for subsequent training in our framework. By leveraging CMD instant performance, we streamline hyperparameter selection and ensure efficient model training.

To summarize, the $M$ (number of modes) and $S$ (number of sampled trajectories) can be efficiently determined by traversing diagonal combinations and comparing the CMD instant performance. This process requires only a single forward pass per combination, making it a highly efficient and practical approach for hyperparameter selection.

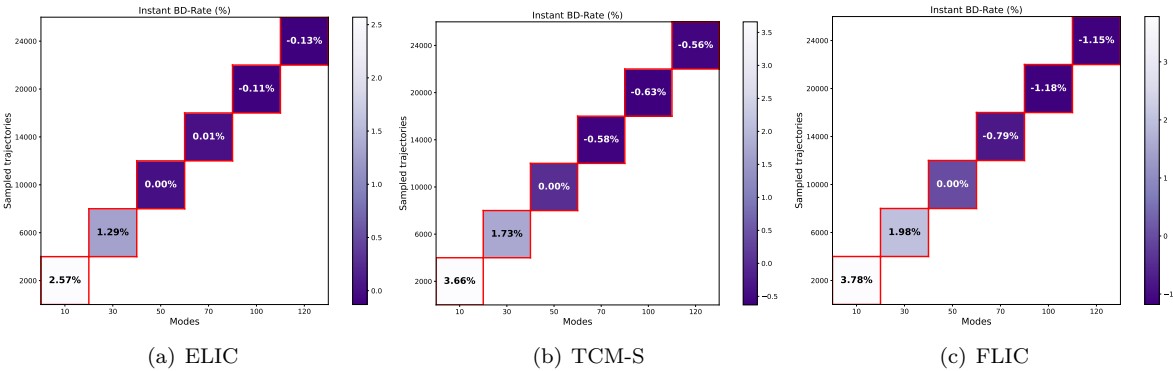

Figure 9: **The instant CMD performance of different modes $M$ and sampled trajectories $S$.** Deeper color represents better performance.

## 5   Conclusion

In this paper, we accelerate the training of LICs by modeling the neural training dynamics, considering the linear representations of LICs' complex models. We extend the concept of Dynamic Mode Decomposition into our proposed Sensitivity-aware True and Dummy Embedding Training (STDET). Our method progressively embeds non-reference parameters based on their stable correlation and sensitivity, thereby reducing the dimension of the training space and the total number of trainable parameters. Additionally, we implement the Sampling-then-Moving Average (SMA) technique to smooth the training trajectory and minimize training variance, enhancing stable correlation. In this way, our approach results in faster convergence and fewer trainable parameters compared to the standard SGD training of LICs across various image domains. Both

analyses on the noisy quadratic model and extensive experimental validations underscore the superiority of our method over standard training schemes for complex, time-intensive LICs.

## 6 Acknowledgment

This work was supported by the National Cancer Institute under Grant R01CA277839.

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

# A    Appendix

This document provides more details about our proposed method, discussion, and comparison.

## A.1    Specific Domain Learned Image Compression Acceleration:

To demonstrate the robustness and real-world applicability of our method, we extend its evaluation to domain-specific scenarios, including stereo image compression, remote sensing image compression, screen content image compression, and raw image compression.

### A.1.1    Stereo Image Compression

We first compare our proposed method with standard SGD-trained models on stereo LICs, specifically ECSIC (Wödlinger et al., 2024) and BiSIC (Liu et al., 2024), to showcase its superior performance. Standard SGD-trained models serve as the anchor for calculating BD-Rate (Bjøntegaard, 2001).

**Training Details.** Following established protocols (Wödlinger et al., 2024; Liu et al., 2024), we use the Cityscapes (Cordts et al., 2016) and InStereo2K (Bao et al., 2020) datasets for training and evaluation.

- **Cityscapes:** This dataset contains 5,000 outdoor stereo image pairs ($2048 \times 1024$ resolution), with 2,975 pairs allocated for training and 1,525 for testing.

- **InStereo2K:** This dataset consists of 2,060 indoor stereo image pairs ($1080 \times 860$ resolution), split into 2,010 pairs for training and 50 for testing.

During training:

- For all models in the "SGD" series, the Adam optimizer ($\beta_1 = 0.9$, $\beta_2 = 0.999$) is used with 450 epochs on Cityscapes and 650 epochs on InStereo2K.

- For all models in the "Proposed" series, the Adam optimizer is also used, but training is limited to 260 epochs on Cityscapes and 380 epochs on InStereo2K.

- A batch size of 8 is used, with a learning rate initialized at $1 \times 10^{-4}$, adjusted via a ReduceLROn-Plateau scheduler (0.5 factor, 20 epochs patience).

**Hyperparameter Selection.**

- Predefined epoch $F = 75$ for Cityscapes and $F = 110$ for InStereo2K.

- Embedding percentage $P = 0.27\%$ for Cityscapes and $P = 0.185\%$ for InStereo2K.

- Embedding period: $L = 1$, moving average factor $\alpha = 0.8$, and optimizer step $l = 5$.

These parameters are determined based on the main experiment settings. For instance, in the main LIC experiments, $F$ is approximately $28.5\%$ ($\frac{20}{70}$) of the total training epochs. Consequently, we set $F = 75 \approx 28.5\% \times 260$ for Cityscapes. For the embedding percentage $P$, in the main experiment, $50\%$ of the parameters are embedded over 50 epochs. Here, $50\%$ of the parameters are embedded within $260 - 75 = 185$ epochs. From Appendix 4.4, we know that the models benefit from smaller embedding percentages and more frequent embeddings. Therefore, we set $P = \frac{50\%}{185} \approx 0.27\%$ with $L = 1$. The number of modes $M$ and the number of sampled trajectories $S$ are determined by traversing the same diagonal set described in Appendix 4.4.4. The resulting hyperparameters are $M = 50$, $S = 10k$ for ECSIC (25.74M) and $M = 100$, $S = 20k$ for BiSIC (78.21M). This dependency on model size underscores the adaptability of our method across different architectures and different domains.

**Results and Analysis.** Fig. 10 presents the R-D curves of both methods, demonstrating that our approach closely aligns with SGD-trained models. Tab. 6 highlights the BD-Rate reductions of our method compared to the SGD anchor, as well as the computational complexity in terms of training time, total trainable parameters, and final trainable parameters. Our method consistently achieves comparable or superior performance across datasets and methods. For example, on Cityscapes, "ECSIC + Proposed" achieves a BD-Rate reduction of $-1.87\%$ compared to "ECSIC + SGD", even outperforming standard SGD in some cases. Additionally, "ECSIC + Proposed" requires only 104 hours of training, which is just $58.75\%$ of the training time for "ECSIC + SGD", highlighting significant efficiency improvements. The total trainable parameters are also significantly reduced. For "ECSIC + SGD", the total trainable parameters are $25.74\text{M} \times 450\,\text{epochs} \times 5\,\text{R-D points} = 57,915\text{M}$. In contrast, for "ECSIC + Proposed": During the first 75 epochs, $100\%$ of parameters $\times 25.74\text{M}$. From epochs 76 to 260, we embed $0.27\%$ of parameters per epoch, progressively reducing the trainable parameters from $100\%$ to $50.32\%$. After calculations across all $\lambda$ values, the total trainable parameters for "ECSIC + Proposed" are reduced to $27,574\text{M}$, approximately $47\%$ of the total trainable parameters required by the SGD method. Similarly, the final trainable parameters after the last epoch are reduced from $25.74\text{M}$ to $12.99\text{M}$. By reducing the number of trainable parameters and leveraging sparse training methods (Zhou et al., 2021b), our approach could further accelerate training, as discussed in Appendix Sec. A.5.

Table 6: Computational Complexity and BD-Rate Compared to SGD for Stereo Image

| Method | | Training time ↓ | Total trainable params ↓ | Final trainable params ↓ | BD-Rate (%) ↓ | |
| --- | --- | --- | --- | --- | --- | --- |
| | | | | | Cityscapes | InStereo2K |
| ECSIC (Wödlinger et al., 2024) | + SGD | 177h | 57,915M | 25.74M | 0% | 0% |
| | + Proposed | **104h** | **27,574M** | **12.99M** | **-1.87%** | **-0.79%** |
| BiSIC (Liu et al., 2024) | + SGD | 207h | 175,972M | 78.21M | 0% | 0% |
| | + Proposed | **122h** | **83,702M** | **39.14M** | **-0.65%** | **-0.62%** |

**Training Conditions:** $1 \times$ Nvidia A40 GPU, AMD EPYC 7662 CPU, 1024GB RAM. **Bold** represents better performance.

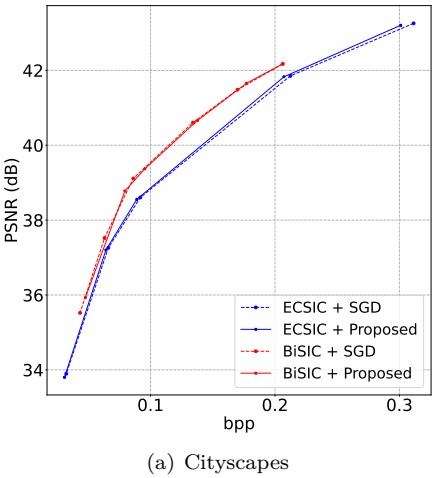 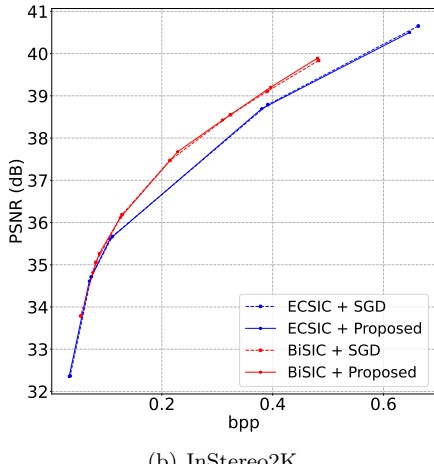

(a) Cityscapes

(b) InStereo2K

Figure 10: **Rate-distortion curves in terms of PSNR on the Cityscapes and InStereo2K datasets.**

### A.1.2 Remote Sensing Image Compression

For remote sensing image compression, we evaluate our method on HL-RSCompNet (Xiang & Liang, 2024) using the following datasets:

- **GEset:** This dataset consists of 8,064 RGB images, each with a size of $640 \times 640$ pixels and a spatial resolution of approximately 0.35 m. Of these, 4,992 images are used for training, and 3,072 are used for testing.

- **GF1set:** The GF1set dataset is derived from the Chinese GaoFen-1 (GF-1) satellite. It comprises 4,560 multispectral images with a spatial resolution of 8 m and an image size of $640 \times 640$ pixels. Among these, 2,400 images are used for training, while the remaining images are reserved for testing. GF1 images include four spectral bands: Band 1 (0.45–0.52 $\mu$m), Band 2 (0.52–0.59 $\mu$m), Band 3 (0.63–0.69 $\mu$m), and Band 4 (0.77–0.89 $\mu$m).

- **GF7set:** The GF7set is obtained from the Chinese GaoFen-7 (GF-7) satellite. It consists of 3,445 images for training and 2,297 images for testing. The images have a size of $640 \times 640$ pixels and a spatial resolution of 3.2 m. GF7 imagery also consists of four spectral bands: Band 1 (0.45–0.52 $\mu$m), Band 2 (0.52–0.59 $\mu$m), Band 3 (0.63–0.69 $\mu$m), and Band 4 (0.77–0.89 $\mu$m).

- **PANset:** The panchromatic dataset is sourced from China's GF-6 satellite. It includes images with a spatial resolution of 2 m and a spectral range of 0.45–0.90 $\mu$m. A total of 2,100 images are used for training, and 1,600 are used for testing. Each image has a size of $512 \times 512$ pixels. This dataset is used to evaluate the compression performance of different methods at higher resolutions.

**Training Configuration.** During training, the following settings are applied:

- For all models in the "SGD" series, the Adam optimizer ($\beta_1 = 0.9$, $\beta_2 = 0.999$) is used, and the models are trained for 600 epochs.

- For all models in the "Proposed" series, the Adam optimizer is also used, but the training is limited to 350 epochs.

- The batch size is set to 8, and the learning rate is initialized at $1 \times 10^{-4}$. A MultiStepLR scheduler with a decay factor of 0.1 and a step size of 200 epochs is employed.

**Hyperparameter Selection.** The hyperparameters for our method are defined as follows:

- Predefined epoch $F = 100$.

- Embedding percentage $P = 0.2\%$.

- Embedding period $L = 1$, moving average factor $\alpha = 0.8$, and optimizer step $l = 5$.

These parameters are also determined based on the main experimental settings. Similar to the previous experiment, the number of modes $M$ and sampled trajectories $S$ are determined by traversing the diagonal set described in Appendix 4.4.4. The resulting hyperparameters for HL-RSCompNet (29.34M) are $M = 50$ and $S = 10k$. This consistent dependence on model size highlights the adaptability of our proposed method across different architectures and domains.

**Results and Analysis.** Fig. 11 shows the R-D curves for both methods, demonstrating that our approach closely matches the performance of SGD-trained models. Tab. 7 provides a detailed comparison of BD-Rate reductions achieved by our method relative to the SGD anchor, along with metrics for computational complexity, including training time, total trainable parameters, and final trainable parameters. Our method consistently delivers comparable or superior performance across all datasets and methods. Additionally, it significantly accelerates the training of HL-RSCompNet, requiring only 60% of the training time while maintaining competitive performance.

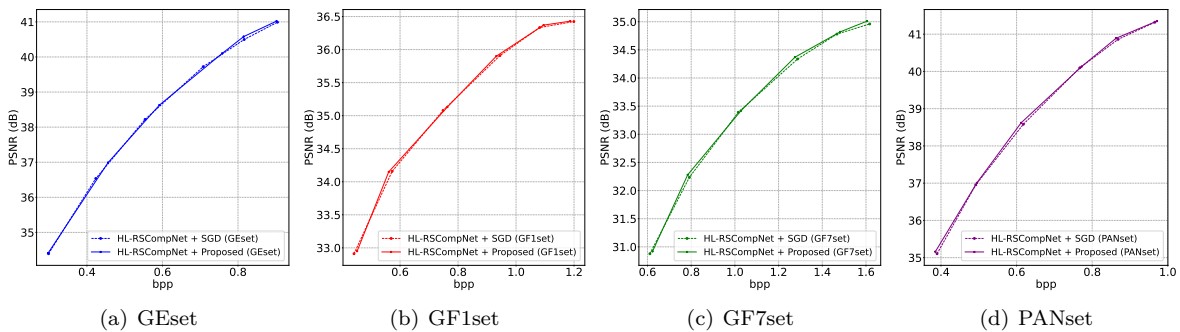

(a) GEset  (b) GF1set  (c) GF7set  (d) PANset

Figure 11: **Rate-distortion curves with different methods on all the datasets.**

Table 7: Computational Complexity and BD-Rate Compared to SGD for Remote Sensing Image

| Method | | Training time ↓ | Total trainable params ↓ | Final trainable params ↓ | BD-Rate (%) ↓ | | | |
|---|---|---|---|---|---|---|---|---|
| | | | | | GEset | GF1set | GF7set | PANset |
| HL-RSCompNet (Xiang & Liang, 2024) | + SGD | 103h | 105,624M | 29.34M | 0% | 0% | 0% | 0% |
| | + Proposed | **62h** | **50,655M** | **14.67M** | **-0.10%** | **-0.59%** | **-0.77%** | **-0.74%** |

**Training Conditions**: $1 \times$ Nvidia A40 GPU, AMD EPYC 7662 CPU, 1024GB RAM. **Bold** represents better performance.

### A.1.3 Screen content image compression

For screen content image compression, we compare our proposed method with standard SGD-trained models on SFTIP (Zhou et al., 2024).

**Training Details.** We follow the configurations specified in the paper (Zhou et al., 2024) for training. The models are trained on the JPEGAI dataset (ISO/IEC JTC 1/SC29/WG1, 2023), which includes 5,264 images for training and 350 images for validation. For evaluation, we use the SIQAD dataset (Yang et al., 2015), comprising 22 high-resolution, high-quality images, and the SCID dataset (Ni et al., 2017), containing 200 high-resolution, high-quality images.

Training Configuration:

- For all models in the "SGD" series, the models are trained using the Adam optimizer for 200 epochs.

- For all models in the "Proposed" series, we also use the Adam optimizer but train for only 120 epochs.

- The batch size is set to 8, and the learning rate is initialized at $1 \times 10^{-4}$, with a ReduceLROnPlateau scheduler (0.3 factor, 4 epochs patience).

The hyperparameters are determined as follows:

- Predefined epoch $F = 36$.

- Embedding percentage $P = 0.6\%$.

- Embedding period $L = 1$, moving average factor $\alpha = 0.8$, and optimizer step $l = 5$.

These parameters are similarly determined using the same process as in previous experiments. The number of modes $M$ and the number of sampled trajectories $S$ are determined by traversing the same diagonal set. The resulting hyperparameters are $M = 100$, $S = 20k$ for SFTIP (95.49M).

**Results and Analysis.** Fig. 12 shows the R-D curves for both methods, demonstrating that our approach closely matches the performance of SGD-trained models. Tab. 8 provides a detailed comparison of BD-Rate reductions achieved by our method relative to the SGD anchor, along with metrics for computational complexity, including training time, total trainable parameters, and final trainable parameters. Our method consistently delivers comparable or superior performance across all datasets and methods. Additionally, it significantly accelerates the training of SFTIP, requiring only 64% of the training time while maintaining competitive performance.

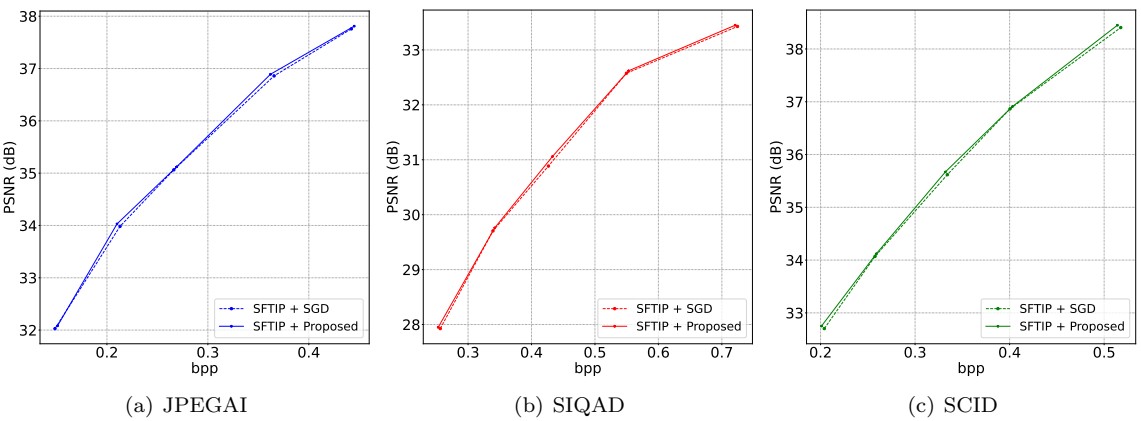

(a) JPEGAI          (b) SIQAD          (c) SCID

Figure 12: **Rate-distortion curves with different compression methods on all the datasets.**

Table 8: Computational Complexity and BD-Rate Compared to SGD for Screen Content Image

| Method | | Training time ↓ | Total trainable params ↓ | Final trainable params ↓ | BD-Rate (%) ↓ | | |
|---|---|---|---|---|---|---|---|
| | | | | | JPEGAI | SIQAD | SCID |
| SFTIP (Zhou et al., 2024) | + SGD | 50h | 95,490M | 95.49M | 0% | 0% | 0% |
| | + Proposed | **32h** | **47,307M** | **47.36M** | **-0.93%** | **-0.77%** | **-0.74%** |

**Training Conditions**: $1 \times$ Nvidia A40 GPU, AMD EPYC 7662 CPU, 1024GB RAM. **Bold** represents better performance.

### A.1.4 RAW image compression

For RAW image compression, we compare our proposed method against standard SGD-trained models on R2LCM (Wang et al., 2024).

**Training Details.** We follow the training configurations specified in (Wang et al., 2024; Nam et al., 2022). The models are trained on the NUS dataset (Cheng et al., 2014), which contains raw images captured by multiple cameras. Specifically, we use three cameras—Samsung NX2000, Olympus E-PL6, and Sony SLT-A57—comprising 202, 208, and 268 raw images, respectively. To preprocess the data, we first demosaic the raw Bayer images using standard bilinear interpolation to obtain 3-channel raw-RGB images. These images are then processed through a software ISP emulator (Karaimer & Brown, 2016) to generate sRGB images, mimicking real-world photo-finishing applied by cameras. The dataset is split into training, validation, and test sets following the protocol in (Nam et al., 2022).

**Training Configuration.** To ensure a fair comparison, both "SGD" and "Proposed" models are trained using the Adam optimizer with the following settings:

- **SGD Series:** Trained for 500 epochs.

- **Proposed Series:** Trained for only 290 epochs, significantly reducing training time.

- **Batch Size:** 8

- **Learning Rate:** Initialized at $1 \times 10^{-4}$ with a ReduceLROnPlateau scheduler (decay factor 0.1, patience of 60 epochs).

**Hyperparameter Selection.** The key hyperparameters are set as follows, using the same selection strategy as in previous experiments:

- Predefined epoch: $F = 80$

- Embedding percentage: $P = 0.24\%$

- Embedding period: $L = 1$, moving average factor $\alpha = 0.8$, and optimizer step $l = 5$.

The number of modes $M$ and the number of sampled trajectories $S$ are determined by traversing the diagonal set, yielding $M = 10$ and $S = 2k$ for R2LCM (2.35M).

**Results and Analysis.** Fig. 13 presents the R-D curves for both methods, showing that our proposed approach achieves performance comparable to that of SGD-trained models. Tab. 9 provides a detailed BD-Rate comparison relative to the SGD baseline, alongside key computational metrics such as training time, total trainable parameters, and final trainable parameters. Our method consistently delivers comparable or superior performance across all datasets and model variants. Notably, it significantly accelerates R2LCM training, requiring only 61% of the training time while maintaining competitive performance. This highlights the effectiveness of our approach in improving training efficiency without compromising model quality on raw image compression task.

Table 9: Computational Complexity and BD-Rate Compared to SGD for RAW Image

| Method | | Training time ↓ | Total trainable params ↓ | Final trainable params ↓ | BD-Rate (%) ↓ | | |
|---|---|---|---|---|---|---|---|
| | | | | | Samsung | Olympus | Sony |
| R2LCM (Wang et al., 2024) | + SGD | 49h | 1175M | 2.35M | 0% | 0% | 0% |
| | + Proposed | **30h** | **569M** | **1.16M** | **-1.16%** | **-0.93%** | **-1.10%** |

**Training Conditions**: 1 × Nvidia A40 GPU, AMD EPYC 7662 CPU, 1024GB RAM. **Bold** represents better performance.

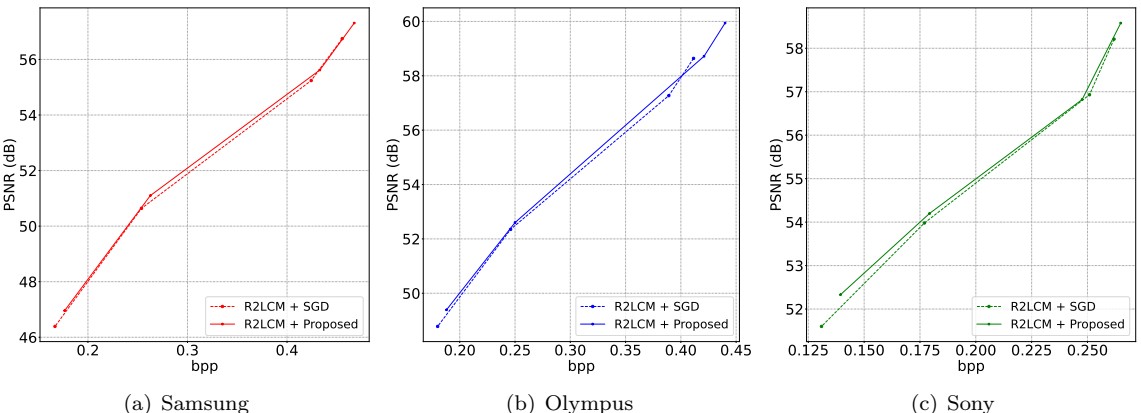

Figure 13: **Rate-distortion curves with different compression methods on all the datasets.**

As has been demonstrated in the extensive experiments on stereo images, remote sensing images, screen content images, and raw images, our method consistently performs effectively across diverse domain-specific image types while adhering to a unified hyperparameter selection strategy. Notably, our approach achieves these results with only 60% of the training time required by standard methods, without any degradation in performance—and in some cases, even showing slight improvements. These findings underscore the capability of our method to significantly accelerate the training process for various image codecs, making it a highly practical and promising solution for real-world applications.

## A.2 Variance reduction

We performed additional experiments to evaluate the variance in model performance across five training runs, comparing the average value and variance of performance between our method and SGD using ELIC (He et al., 2022) as a reference. Specifically, we trained the ELIC model with all $\lambda$s five times using both our method and SGD. The anchor is VVC-intra with a yuv444 config following standard practie (He et al., 2022; Liu et al., 2023; Li et al., 2024a; Zhang et al., 2024b). As shown in Tab. 10, our method achieves both lower average BD-Rate and reduced variance compared to SGD. These results are consistent with our theoretical analysis, further underscoring the stability of the proposed method.

Table 10: Performance comparison across five runs.

| Method | Avg BD-Rate (Variance) |
|---|---|
| ELIC + SGD | -5.83% (0.0184) |
| ELIC + Proposed | -6.53% (0.0023) |

To gain deeper insights into the variance behavior of the training process, we present the mean loss curve for $\lambda = 0.0018$ along with its standard deviation[5] range, as shown in Fig. 15. The solid line represents the mean loss computed over five independent runs at each training epoch, while the shaded region denotes the corresponding standard deviation, capturing the variability in training dynamics. As observed, the variance of standard SGD is significantly larger than that of the proposed method. This observation is consistent with the BD-Rate results in Tab. 10 and the theoretical analysis in Sec. A.6, further demonstrating the low-variance property of our proposed approach.

## A.3 Discussion on STDET and SMA

To better understand the distinct contributions of each proposed technique, we conducted additional experiments to evaluate them independently. The experimental settings were kept identical to those used in

---

[5]Standard deviation is more observable due to the relatively small value of variance

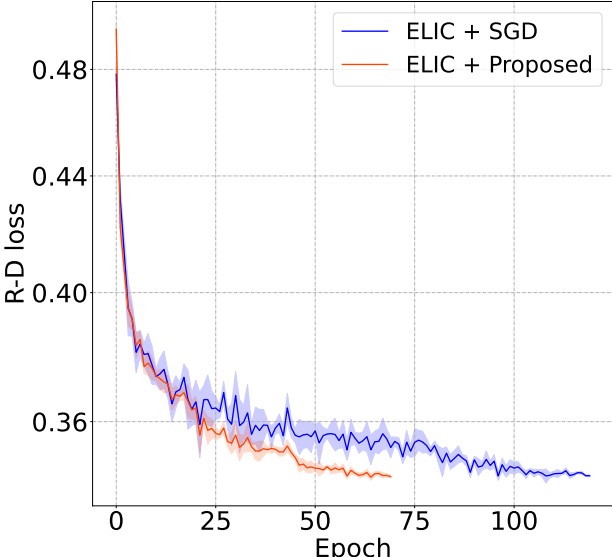

Figure 14: Testing loss behavior across five independent runs for different methods. The evaluation is conducted on the Kodak dataset with $\lambda = 0.0018$. The testing R-D loss is computed as $\lambda \cdot 255^2 \cdot \mathrm{MSE} + \mathrm{Bpp}$.

the main experiments. We employed the ELIC model (He et al., 2022) with $\lambda = 0.0018$ as the illustrative baseline. The results are depicted in Fig. 15.

First, we evaluated STDET without incorporating SMA (ELIC + STDET). When applied in isolation, STDET struggled to deliver a competitive performance. This limitation arises from the stochastic nature of the training process, which introduces variability and affects the consistency and accuracy of STDET's correlation modeling. Consequently, the inaccuracies in parameter embedding lead to a noticeable performance drop. This highlights the need for complementary techniques to smooth and stabilize the training trajectories.

Next, we applied SMA independently to a standard SGD-trained LIC model (ELIC + SMA). The results demonstrate that SMA achieves R-D performance comparable to both standard SGD and the final combined setup ("STDET + SMA"), with the added benefit of a smoother training trajectory. However, it does not accelerate the training process. This confirms that SMA primarily acts as a smoothing and stabilizing technique when used alone and does not directly contribute to additional performance improvements or much faster convergence, as anticipated.

Overall, the combination of STDET and SMA offers the best balance between training efficiency and performance. STDET reduces the dimensionality of the training process, thereby accelerating training under stable correlation modeling, while SMA ensures stability throughout the process, enabling accurate correlation modeling, which is required by STDET. Together, these techniques enable an efficient and accurate acceleration of the LIC training process.

### A.4 Comparison of EMA, explicit alignment and SMA

To clarify the differences between EMA, explicit alignment, and SMA, Fig. 16 provides an illustrative comparison of the three methods over four training iterations.

**For EMA,** In each training iteration, we first perform a standard SGD update on the model parameters $w_{SGD}$. Then, we update the EMA parameters $w_{EMA}$ using the newly obtained $w_{SGD}$ and the previous $w_{EMA}$ state. Notably, during training, $w_{EMA}$ does not influence $w_{SGD}$—it is updated passively and used only for testing after training ends. The trajectory of $w_{SGD}$ remains noisy, while $w_{EMA}$ follows a smoother trajectory.

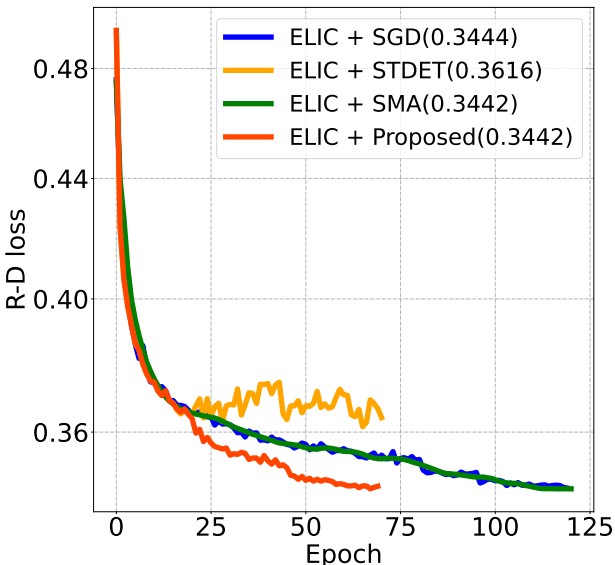

Figure 15: Testing loss comparison of various techniques. The upper right corner shows the final convergence result. $\lambda = 0.0018$, Testing R-D loss $= \lambda \cdot 255^2 \cdot \mathrm{MSE} + \mathrm{Bpp}$. Evaluated on the Kodak dataset.

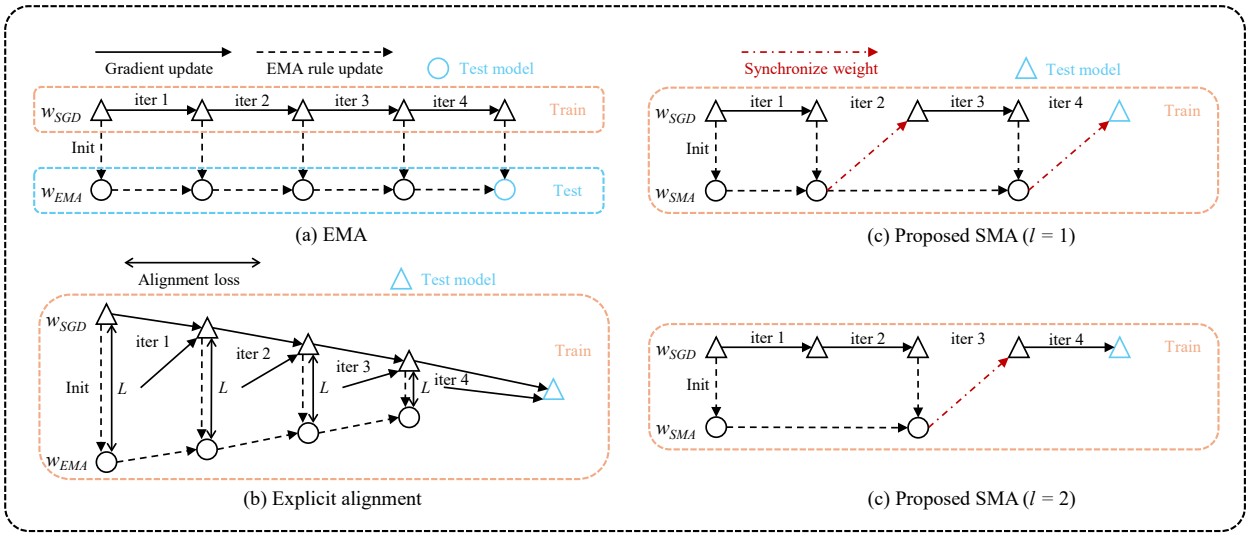

Figure 16: Illustrative comparison of EMA (Szegedy et al., 2016; Polyak & Juditsky, 1992), explicit alignment (Song et al., 2023), and SMA ($l = 1$ & $l = 2$) over four training iterations.

**For explicit alignment,** the process begins similarly: $w_{SGD}$ is updated through standard SGD, and $w_{EMA}$ is computed following the EMA update rule. However, an additional loss function is introduced to align the SGD trajectory with the EMA trajectory explicitly. This loss is computed based on the discrepancy between the outputs of the current SGD model and the EMA model. By minimizing this loss, $w_{SGD}$ is encouraged to follow a smoother trajectory that aligns with $w_{EMA}$. After sufficient training, $w_{SGD}$ and $w_{EMA}$ ideally converge to the same weight, leading to a stabilized final model. However, this method introduces extra computational overhead due to the additional loss calculation and gradient updates.

**For SMA**, we periodically synchronize the EMA-updated parameters at sampled intervals to the $w_{SGD}$.

- Taking $l = 1$ as an example:
    - In the first iteration, we perform a standard SGD update on $w_{\text{SGD}}$. After the update, we sample this iteration's parameters and update $w_{\text{SMA}}$ using the EMA update rule.
    - Once $w_{\text{SMA}}$ is updated, we synchronize it back to $w_{\text{SGD}}$, and the synchronized $w_{\text{SGD}}$ is used as the starting point for the next iteration. This process is seen as the second iteration.
    - In the third iteration, another normal SGD update is performed, followed by an update to $w_{\text{SMA}}$ in the fourth iteration, where synchronization occurs again.
    - This pattern continues throughout training.

- For the case of $l = 2$, the process remains the same, except that two consecutive SGD iterations occur before updating $w_{SMA}$. The next synchronization step then serves as an iteration. This pattern repeats, alternating between two SGD iterations and an SMA update with synchronization.

In essence, our approach periodically samples parameters from the SGD trajectory, applies an EMA-style update to obtain $w_{SMA}$, and synchronizes it back to $w_{SGD}$ before continuing training. Since $w_{SMA}$ follows an EMA-style update, it naturally forms a smoother trajectory, making the final training trajectory more stable.

In summary, both explicit alignment and SMA integrate EMA into the training phase to enhance stability. However, SMA achieves this in a more computationally efficient manner, as it does not have additional loss computations and gradient updates. Instead, it interpolates sampled SGD parameters and synchronizes them with $w_{SGD}$, offering a simple yet effective way to smooth the training process.

### A.5 Limitations and future work.

**Additional Memory Consumption:** The proposed method requires more memory due to additional non-learnable parameters. The notable additional parameters and their relative sizes are detailed below:

- STDET: Vectors $K$ and $D$, each with a size equal to that of the model parameters.

- STDET: A vector storing the mode related to each weight, with the same length as $K$ but containing positive integers representing mode numbers. This vector's size is approximately 0.25 times that of the model.

- SMA: SMA model parameters $w_{\text{SMA}}$, with a size equal to that of the model parameters.

Note that the GPU memory used to store these parameters is insignificant during the training process compared to the sustained gradients (Izmailov et al., 2018), making the overall increase negligible.

**Integration of Sparse Training:** With the embedding of LICs' parameters, they cease to be learnable, meaning that in practice, we can avoid the time-consuming gradient computations during backpropagation for these parameters. However, due to the constraints of existing platforms like TensorFlow and PyTorch, this is not straightforwardly achievable. These platforms rely on the chain rule to compute gradients, and simply setting `requires_grad=False` does not prevent the computation of these parameters within the chain rule; it only avoids the storage of their gradients. A potential solution could involve integrating

the proposed approach with sparse training techniques (Zhou et al., 2021b), thereby completely bypassing gradient computations for these parameters during each training iteration. As a result, with an increasing number of embedded parameters, the training time per epoch would decrease, further accelerating the overall training process. Nevertheless, implementing such an optimization within current frameworks is non-trivial and beyond the scope of this work, leaving it as a promising direction for future research.

**Broader Application to Learned Video Compression:** Our proposed method has been shown to effectively accelerate a wide range of LICs across diverse domain images through both theoretical analysis and extensive experiments. A natural extension would be its application to learned video compression (LVC). Notably, the intra-frame coding component of LVC models and the contextual coders in the DCVC series (Li et al., 2021; Sheng et al., 2022; Li et al., 2022a; 2023a; 2024b; Jia et al., 2025) are fundamentally LICs. Therefore, our method could be utilized to directly accelerate the training of these frame coders. However, the motion estimation and temporal modeling components in DCVCs introduce additional complexities that require further investigation. Exploring ways to extend our acceleration approach to these components remains an open challenge for future work.

## A.6 Noisy quadratic analysis

In this section, we analyze the proposed method on a noisy quadratic model. The quadratic noise function is a commonly adopted base model for analyzing the optimization process, where the stochasticity incorporates noise introduced by mini-batch sampling (Schaul et al., 2013; Wu et al., 2018; Zhang et al., 2019b; Koh & Liang, 2017). We will show that the proposed method converges to a smaller steady-state training variance than the normal SGD training.

**Model Definition.** The quadratic noise model is defined as:

$$\hat{L}(\theta) = \frac{1}{2}(\theta - \mathbf{c})^T \mathbf{A}(\theta - \mathbf{c}), \tag{11}$$

where $\mathbf{c} \sim \mathcal{N}(\theta^*, \Sigma)$, $\mathbf{A}$ and $\Sigma$ are diagonal matrices, and $\theta^* = 0$ (Wu et al., 2018; Zhang et al., 2019a; 2023). Let $a_i$ and $\sigma_i^2$ be the $i$-th elements on the diagonal of $\mathbf{A}$ and $\Sigma$, respectively. The expected loss of the iterates $\theta_t$ can be written as:

$$\mathcal{L}(\theta_t) = \mathbb{E}[\hat{L}(\theta_t)] = \frac{1}{2}\sum_i a_i(\mathbb{E}[\theta_{t,i}]^2 + \mathbb{V}[\theta_{t,i}] + \sigma_i^2), \tag{12}$$

where $\theta_{t,i}$ represents the $i$-th element of the parameter $\theta_t$. The steady-state risk of SGD and the proposed method are compared by unwrapping $\mathbb{E}[\theta_t]$ and $\mathbb{V}[\theta_t]$ in Eq. 12. The expectation trajectories $\mathbb{E}[\theta_t]$ contract to zero in both SGD and the proposed method. Thus, we focus on the variance:

**Steady-state risk.** Let $0 < \gamma < 1/L$ be the learning rate satisfying $L = \max_i a_i$. In the noisy quadratic setup, the variance of the iterates obtained by SGD and the proposed method converges to the following matrices:

$$V_{\text{SGD}}^* = \frac{\gamma^2 \mathbf{A}^2 \Sigma}{\mathbf{I} - (\mathbf{I} - \gamma\mathbf{A})^2}, \tag{13}$$

$$V_{\text{Proposed}}^* = \underbrace{\frac{(1 - p + p\mathbb{E}[k_i^2])\alpha^2(\mathbf{I} - (\mathbf{I} - \gamma\mathbf{A})^{2l})}{\alpha^2(\mathbf{I} - (\mathbf{I} - \gamma\mathbf{A})^{2l}) + 2\alpha(1 - \alpha)(\mathbf{I} - (\mathbf{I} - \gamma\mathbf{A})^l)}}_{\leq \mathbf{I}, \text{ if } \alpha \in (0,1) \text{ and } \mathbb{E}[k_i^2] \leq 1} V_{\text{SGD}}^*, \tag{14}$$

where $\alpha$ denotes the SMA weighting factor with varying trajectory states, $p$ is the percentage of final embedded parameters, and $\mathbb{E}[k_i^2]$ is the expectation of squared affine coefficients $k_i^2$.

From Eq. 14, it is evident that for the steady-state variance of the proposed method, the first product term is always smaller than $\mathbf{I}$ for $\alpha \in (0,1)$ and $\mathbb{E}[k_i^2] \leq 1$. Therefore, the proposed method exhibits a smaller steady-state variance than the SGD optimizer at the same learning rate, resulting in a lower expected loss (see Eq. 12). Additionally, due to the reduced variance in the training process, the correlation within the

same modes is highly preserved, ensuring that the STDET avoids serious failure cases. The small steady-state variance ensures good generalizability in real-world scenarios (Wu et al., 2018) and demonstrates why the proposed method converges well.

***Proof.*** Here we present the proof of Eq. 14. We use $\theta$ to represent normal SGD training parameters and $\phi$ to represent SMA-updated training parameters.

**Stochastic dynamics of SGD:** From Wu et al. (2018), we can obtain the dynamics of SGD with learning rate $\gamma$ as follows:

$$\mathbb{E}[\mathbf{x}^{(t+1)}] = (\mathbf{I} - \gamma\mathbf{A})\mathbb{E}[\mathbf{x}^{(t)}], \tag{15}$$

$$\mathbb{V}[\mathbf{x}^{(t+1)}] = (\mathbf{I} - \gamma\mathbf{A})^2\mathbb{V}[\mathbf{x}^{(t)}] + \gamma^2\mathbf{A}^2\Sigma. \tag{16}$$

**Stochastic dynamics of Proposed method:** We now compute the dynamics of the proposed method. The expectation and variance of the SMA have the following iterates based on the lookahead optimizers (Zhang et al., 2019b; 2023):

The expectation trajectory is represented as:

$$\begin{aligned}
\mathbb{E}[\phi_{t+1}] &= (1-\alpha)\mathbb{E}[\phi_t] + \alpha\mathbb{E}[\theta_{t,l}] \\
&= (1-\alpha)\mathbb{E}[\phi_t] + \alpha(\mathbf{I} - \gamma\mathbf{A})^l\mathbb{E}[\phi_t] \\
&= \left[1 - \alpha + \alpha(\mathbf{I} - \gamma\mathbf{A})^l\right]\mathbb{E}[\phi_t].
\end{aligned} \tag{17}$$

When combined with STDET, in the training dynamics, although the embedded parameters are not trainable, they depend on the reference parameters, which are also updated by SMA. Thus, all the parameters follow the same expectation dynamics.

For the variance, we can write

$$\mathbb{V}[\phi_{t+1}] = (1-\alpha)^2\mathbb{V}[\phi_t] + \alpha^2\mathbb{V}[\theta_{t,l}] + 2\alpha(1-\alpha)\text{cov}(\phi_t, \theta_{t,l}). \tag{18}$$

Also, it's easy to show:

$$\mathbb{V}[\theta_{t,l}] = (\mathbf{I} - \gamma\mathbf{A})^{2l}\mathbb{V}[\phi_t] + \gamma^2\sum_{i=0}^{l-1}(\mathbf{I} - \gamma\mathbf{A})^{2i}\mathbf{A}^2\Sigma,$$

$$\text{cov}(\phi_t, \theta_{t,l}) = (\mathbf{I} - \gamma\mathbf{A})^l\mathbb{V}[\phi_t]. \tag{19}$$

After substituting the SGD variance formula and some rearranging, we have:

$$\mathbb{V}[\phi_{t+1}] = \left[1 - \alpha + \alpha(\mathbf{I} - \gamma\mathbf{A})^l\right]^2\mathbb{V}[\phi_t] + \alpha^2\sum_{i=0}^{l-1}(\mathbf{I} - \gamma\mathbf{A})^{2i}\gamma^2\mathbf{A}^2\Sigma. \tag{20}$$

Similarly, for the variance term, all the parameters follow the same SMA dynamics. Then, for the proposed method:

$$\mathbb{E}[\phi_{t+1}] = \left[1 - \alpha + \alpha(\mathbf{I} - \gamma\mathbf{A})^l\right]\mathbb{E}[\phi_t], \tag{21}$$

$$\mathbb{V}[\phi_{t+1}] = \left[1 - \alpha + \alpha(\mathbf{I} - \gamma\mathbf{A})^l\right]^2\mathbb{V}[\phi_t] + \alpha^2\sum_{i=0}^{l-1}(\mathbf{I} - \gamma\mathbf{A})^{2i}\gamma^2\mathbf{A}^2\Sigma. \tag{22}$$

Now, we proceed to find the steady state of the expectation and variance:

The given expectation term is:

$$\mathbb{E}[\phi_{t+1}] = \left[1 - \alpha + \alpha(\mathbf{I} - \gamma\mathbf{A})^l\right]\mathbb{E}[\phi_t]. \tag{23}$$

Define the mapping $T$ as:

$$T(\phi_t) = \left[1 - \alpha + \alpha(\mathbf{I} - \gamma\mathbf{A})^l\right]\phi_t. \tag{24}$$

A mapping $T$ is a contraction if there exists a constant $0 \leq c < 1$ such that for all $\phi_t$ and $\phi_t'$:

$$\|T(\phi_t) - T(\phi_t')\| \leq c\|\phi_t - \phi_t'\|. \tag{25}$$

In our case:
$$\begin{aligned}
\|T(\phi_t) - T(\phi_t')\| &= \left\| \left[1 - \alpha + \alpha(\mathbf{I} - \gamma\mathbf{A})^l\right]\phi_t - \left[1 - \alpha + \alpha(\mathbf{I} - \gamma\mathbf{A})^l\right]\phi_t' \right\| \\
&= \left\| \left[1 - \alpha + \alpha(\mathbf{I} - \gamma\mathbf{A})^l\right](\phi_t - \phi_t') \right\|.
\end{aligned} \tag{26}$$

Let $M = 1 - \alpha + \alpha(\mathbf{I} - \gamma\mathbf{A})^l$. We need to find the spectral radius $\rho(M)$, which represents the largest absolute value of the eigenvalues of $M$. Since $(\mathbf{I} - \gamma\mathbf{A})$ is a contraction matrix (assuming the learning rate $\gamma$ is chosen such that $0 < \gamma < 1/L$, where $L = \max_i a_i$), its eigenvalues $\lambda_i$ satisfy:

$$|\lambda_i| < 1. \tag{27}$$

Therefore, the eigenvalues of $(\mathbf{I} - \gamma\mathbf{A})^l$ are $\lambda_i^l$ and satisfy:

$$|\lambda_i^l| < 1. \tag{28}$$

Next, consider the matrix $M$:

$$M = 1 - \alpha + \alpha(\mathbf{I} - \gamma\mathbf{A})^l. \tag{29}$$

The eigenvalues of $M$ are given by:

$$\rho(M) = \left|1 - \alpha + \alpha\lambda_i^l\right|. \tag{30}$$

For the mapping $T$ to be a contraction, the spectral radius $\rho(M)$ must be less than 1:

$$\left|1 - \alpha + \alpha\lambda_i^l\right| < 1 \tag{31}$$

Given $0 < \alpha < 1$ and $|\lambda_i^l| < 1$, the inequality $\left|1 - \alpha + \alpha\lambda_i^l\right| < 1$ will hold. By Banach's Fixed Point Theorem (Oltra & Valero, 2004), since $T$ is a contraction mapping, there exists a unique fixed point $E^*$ such that $T(E^*) = E^*$.

To find the fixed point, we set $\mathbb{E}[\phi_{t+1}] = \mathbb{E}[\phi_t] = E^*$:

$$E^* = \left[1 - \alpha + \alpha(\mathbf{I} - \gamma\mathbf{A})^l\right]E^*. \tag{32}$$

Simplifying:

$$\left[\alpha - \alpha(\mathbf{I} - \gamma\mathbf{A})^l\right]E^* = 0. \tag{33}$$

Since $\alpha \neq 0$. The only solution is:

$$E^* = 0. \tag{34}$$

Using the contraction mapping approach, we have shown that the expectation term indeed converges to the fixed point $E^* = 0$.

Now, we need to consider the effect of STDET. Let $p$ represent the percentage of final embedded parameters. Thus $E^* = pE^*_{\text{embed}} + (1-p)E^*_{\neg\text{embed}}$, where $E^*_{\text{embed}} \to 0$ and $E^*_{\neg\text{embed}} = \mathbb{E}[k_i]E^*_{\text{embed}} + \mathbb{E}[d_i] \to \mathbb{E}[d_i]$, where we empirically find that $\mathbb{E}[d_i] \to 0$ in our cases. Thus, overall, the expectation trajectories contract to zero. Clearly, the expectation trajectories of SGD also contract to zero as it is also a contract map.

Similarly, for the variance, under the same condition, we know that:

$$\begin{aligned}
V^*_{\text{SGD}} &= (1 - \gamma\mathbf{A})^2 V^*_{\text{SGD}} + \gamma^2\mathbf{A}^2\Sigma \\
&= \frac{\gamma^2\mathbf{A}^2\Sigma}{\mathbf{I} - (\mathbf{I} - \gamma\mathbf{A})^2}.
\end{aligned} \tag{35}$$

For the proposed method,

$$
\begin{aligned}
V_{\text{SMA}}^* &= \left[1 - \alpha + \alpha(\mathbf{I} - \gamma\mathbf{A})^l\right]^2 V_{\text{SMA}}^* + \alpha^2 \sum_{i=0}^{l-1}(\mathbf{I} - \gamma\mathbf{A})^{2i}\gamma^2\mathbf{A}^2\Sigma \\
&= \frac{\alpha^2 \sum_{i=0}^{l-1}(\mathbf{I} - \gamma\mathbf{A})^{2i}}{\mathbf{I} - \left[(1-\alpha)\mathbf{I} + \alpha(\mathbf{I} - \gamma\mathbf{A})^l\right]^2}\gamma^2\mathbf{A}^2\Sigma \\
&= \frac{\alpha^2(\mathbf{I} - (\mathbf{I} - \gamma\mathbf{A})^{2l})}{\mathbf{I} - \left[(1-\alpha)\mathbf{I} + \alpha(\mathbf{I} - \gamma\mathbf{A})^l\right]^2}\frac{\gamma^2\mathbf{A}^2\Sigma}{\mathbf{I} - (\mathbf{I} - \gamma\mathbf{A})^2} \\
&= \frac{\alpha^2(\mathbf{I} - (\mathbf{I} - \gamma\mathbf{A})^{2l})}{\alpha^2(\mathbf{I} - (\mathbf{I} - \gamma\mathbf{A})^{2l}) + 2\alpha(1-\alpha)(\mathbf{I} - (\mathbf{I} - \gamma\mathbf{A})^l)}\frac{\gamma^2\mathbf{A}^2\Sigma}{\mathbf{I} - (\mathbf{I} - \gamma\mathbf{A})^2} \\
&= \frac{\alpha^2(\mathbf{I} - (\mathbf{I} - \gamma\mathbf{A})^{2l})}{\alpha^2(\mathbf{I} - (\mathbf{I} - \gamma\mathbf{A})^{2l}) + 2\alpha(1-\alpha)(\mathbf{I} - (\mathbf{I} - \gamma\mathbf{A})^l)}V_{\text{SGD}}^*.
\end{aligned}
\tag{36}
$$

Also, together with the STDET:

$$
\begin{aligned}
V_{\text{Proposed}}^* &= pV_{\text{embed}}^* + (1-p)V_{\neg\text{embed}}^* \\
&= p\mathbb{E}[k_i^2])V_{\text{SMA}}^* + (1-p)V_{\text{SMA}}^* \\
&= \underbrace{\frac{(1 - p + p\mathbb{E}[k_i^2])\alpha^2(\mathbf{I} - (\mathbf{I} - \gamma\mathbf{A})^{2l})}{\alpha^2(\mathbf{I} - (\mathbf{I} - \gamma\mathbf{A})^{2l}) + 2\alpha(1-\alpha)(\mathbf{I} - (\mathbf{I} - \gamma\mathbf{A})^l)}}_{\leq\mathbf{I},\text{ if }\alpha\in(0,1)\text{ and }\mathbb{E}[k_i^2]\leq 1}V_{\text{SGD}}^*,
\end{aligned}
\tag{37}
$$

Thus, we have shown Eq. 14, and the proposed method will achieve a smaller loss for the same learning rate as the variance is reduced in Eq. 12.

### A.7 Detailed algorithm

The full practical STDET algorithm is presented in Algorithm 1.

### A.8 Experiments details

#### A.8.1 Detailed Training Settings

We list detailed training information in Tab 11, including data augmentation, hyperparameters, and training devices.

#### A.8.2 Used implementations

We list the implementations of the learning-based image codecs that we used for comparison in Tab. 12. We list the compared efficient training method implementations in Tab. 13

Table 11: Training Hyperparameters.

| | |
|---|---|
| Training set | COCO 2017 train |
| # images | 118,287 |
| Image size | Around 640x420 |
| Data augmentation | Crop, h-flip |
| Train input size | 256x256 |
| Optimizer | Adam |
| Learning rate | $1 \times 10^{-4}$ |
| LR schedule | ReduceLROnPlateau |
| LR schedule parameters | factor: 0.5, patience: 5 |
| Batch size | 16 |
| Epochs (SGD) | 120 ($\lambda =$0.0018), 80 (others) |
| Epochs (Proposed) | 70 ($\lambda =$0.0018), 50 (others) |
| Gradient clip | 2.0 |
| GPUs | $1 \times$ RTX 4090 |
| STDET | |
| Predefined epoch $F$ | 20 ($\lambda =$0.0018), 10 (others) |
| Embeddable parameters | 75% |
| Embedding period $L$ | 1 |
| True embedding percentage $P$ | 1% |
| Dummy embedding percentage | $\min\left(\frac{t}{2}\%, 25\%\right)$, where $t$ is the current epoch |
| SMA | |
| Moving average factors $\alpha$ | 0.8 |
| Optimizer step $l$ | 5 |

Table 12: Learning-based Codecs Implementations.

| Method | Implementation |
|---|---|
| ELIC (He et al., 2022) | `https://github.com/InterDigitalInc/CompressAI` |
| TCM-S (Liu et al., 2023) | `https://github.com/jmliu206/LIC_TCM` |
| FLIC (Li et al., 2024a) | `https://github.com/qingshi9974/ICLR2024-FTIC` |
| ECSIC (Wödlinger et al., 2024) | `https://github.com/mwoedlinger/ecsic` |
| BiSIC (Liu et al., 2024) | `https://github.com/LIUZhening111/BiSIC` |
| HL-RSCompNe (Xiang & Liang, 2024) | `https://github.com/shao15xiang/HL-RSCompNet` |
| SFTIP (Zhou et al., 2024) | `https://github.com/vpaHduGroup/SFTIP_SCC` |
| R2LCM (Wang et al., 2024) | `https://github.com/wyf0912/R2LCM` |

Table 13: Efficient Training Methods Implementations.

| Method | Implementation |
|---|---|
| P-SGD (Li et al., 2022b) | `https://github.com/nblt/DLDR` |
| P-BFGS (Li et al., 2022b) | `https://github.com/nblt/DLDR` |
| TWA (Li et al., 2023b) | `https://github.com/nblt/TWA` |
| RigL (Evci et al., 2020) | `https://github.com/google-research/rigl` |
| SRigL (Lasby et al., 2024) | `https://github.com/calgaryml/condensed-sparsity` |

---

**Algorithm 1** STDET Algorithm

---

**Hyper-parameters:**

- $F$ (Predefined epochs), $L$ (Embedding period), $P$ (Embedding percentage), $M$ (Number of modes) range, $S = 200M$ (Number of sampled trajectories) range, Other inputs required for the training process and STDET (learning rate, etc.)

**Procedure:**

1. Run $F$ regular SGD epochs to obtain weight trajectories $W_F$.

2. Initialize variables: prev R-D loss $\leftarrow 10,000$.

3. **for each $M, S = 200M$ within the pre-defined range do**

   (a) Uniformly sample $S$ trajectories from $\mathbf{W}_F$ to avoid calculating a large $N \times N$ correlation matrix.

   (b) Calculate the sampled trajectory correlation matrix $C$:
   $$C[i,j] = \left| \text{corr}(w_{i,F}, w_{j,F}) \right|, \quad C \in \mathbb{R}^{S \times S}.$$

   (c) Cluster the sampled trajectories into $M$ modes using the Farthest Point hierarchical clustering method (Müllner, 2011).

   (d) Select $w_{m,F}$ as the parameter trajectory with the highest average correlation to the other parameters within each mode.

   (e) Calculate the remaining (not sampled) trajectory correlations:
   $$C_2[i,m] = \left| \text{corr}(w_{i,F}, w_{m,F}) \right|, \quad C_2 \in \mathbb{R}^{(N-S) \times M}.$$

   (f) Assign each $w_{i,F} \in \mathbf{W}_F$ to a mode by selecting the arg max of the $i$-th row of $C_2[i,:]$.

   (g) Compute $\tilde{K}_m(F) = \left[ K_m^T, D_m^T \right]$ for each mode using the pseudo-inverse of Eq. 4:
   $$\tilde{K}_m(F) = \mathbf{W}_{m,F} \tilde{w}_{m,F}^T (\tilde{w}_{m,F} \tilde{w}_{m,F}^T)^{-1},$$
   $$\in \arg \min_{\tilde{K}_m(F)} \| \mathbf{W}_{m,F} - \tilde{K}_m \tilde{w}_{m,F} \|_{\text{FN}},$$

   where $\| \cdot \|_{\text{FN}}$ denotes the Frobenius norm, $\tilde{w}_m = \begin{bmatrix} w_m \\ \vec{1} \end{bmatrix}$, and $\vec{1} = (1, 1, \ldots, 1)$.

   (h) Get the CMD instant model by Eq. 4, and evaluate CMD instant performance for the current $M, S$ on the sampled dataset, calculate R-D loss$_{\text{current}}$.

   (i) **Check for saturation:**
   $$\Delta_{\text{R-D}} = \frac{|\text{R-D loss}_{\text{current}} - \text{prev R-D loss}|}{\text{prev R-D loss}}$$

   **if $\Delta_{\text{R-D}} < 0.001$ then**
   - Keep current $K$, $D$, and mode decomposition results.
   - **Break loop.**

   (j) **Update:** prev R-D loss $\leftarrow$ R-D loss$_{\text{current}}$.

4. **end for**

5. Find embeddable parameters by calculating the parameter sensitivity on the sampled dataset.

6. Initialize the set of embedded weights: $\mathcal{I} \leftarrow \emptyset$.

7. **for $t > F$ do**

   (a) Perform a regular SGD epoch.

   (b) Iteratively update $K$ and $D$ using Eq. 6.

   (c) Update weights $w_i(t)$ as follows:
   $$w_i(t) \leftarrow \begin{cases} k_i w_r(t) + d_i & \text{if } i \in \mathcal{I} \\ \text{SGD update} & \text{if } i \notin \mathcal{I} \end{cases}$$

   (d) **if $(t - F)\%L == 0$ then**

      i. Compute the long-term change:
      $$C(t) = \sqrt{(K(t) - K(t-L))^2 + (D(t) - D(t-L))^2}$$

      ii. Update $\mathcal{I}$ to include the weights with the least $P\%$ changes in terms of $C$, excluding reference weights and already embedded weights.

      iii. Reinitialize additional $\min \left( \frac{tP}{2}\%, 25\% \right)$ parameters that have the least $C$ changes excluding $\mathcal{I}$ by: $w_i \leftarrow k_i w_m(t) + d_i$.

   (e) **end if**

8. **end for**

---

