# OpenReview forum: "Accelerating Learned Image Compression Through Modeling Neural Training Dynamics"
_TMLR — Accepted by TMLR_

### Review · Reviewer_jsnA · 2025-02-26

**Summary Of Contributions:**

This paper introduces a novel method for accelerating the training of existing learned image compression models. It proposes a sensitivity-aware true and dummy embedding training mechanism to achieve a linear approximation of complex nonlinear learned image compressors (LICs). Additionally, it integrates a sampling-then-moving average principle to update both reference and non-reference parameters. Experimental results indicate that the proposed method not only reduces training time but also slightly enhances compression performance across various advanced baselines.

**Audience:**

Yes

**Broader Impact Concerns:**

I think there is not ethical implications of the work.

**Claims And Evidence:**

Yes

**Requested Changes:**

Is it possible to apply the proposed method to the latest neural video compression techniques, such as the DCVC codec series [1]? Please discuss this in the appendix.

[1] https://github.com/microsoft/DCVC

**Strengths And Weaknesses:**

Strengths:
1. Introduce a novel STDET mechanism to approximate the stochastic gradient descent training of learned image codecs within a low-dimensional space.
2. Provide a theoretical analysis using the noisy quadratic model, demonstrating that the proposed method achieves lower training variance compared to standard SGD.
3. Achieve faster convergence speed without compromising compression performance.
4. Conduct extensive studies on various types of image compression models.

Weaknesses:
The paper could benefit from a discussion on future work, particularly regarding the application of the proposed method to other neural compression domains, such as video codecs. Exploring the adaptability and effectiveness of this approach across different types of neural compression could provide valuable insights and broaden its potential impact.

---

> ### Author Response · Authors · 2025-03-14
> **Response to Reviewer jsnA: New discussion on video compression.**
>
> Dear Reviewer jsnA,
>
> Thank you for your valuable feedback and for recognizing the novel, significance of our work and comprehensive experiments.
>
> **1. discussion on video compression**
>
> We thank the reviewer for the time and thoughtful feedback. Yes, we believe our method has the potential to be applied to video compression, such as DCVC codecs. However, due to the unavailability of the DCVCs training script at this moment, we are currently unable to conduct experiments on this. Nevertheless, we provide a discussion on this in Section A.5 (Limitations and Future Work – Broader Application to Learned Video Compression). For your convenience, we have copied it here:
>
> ""Our proposed method has been shown to effectively accelerate a wide range of LICs across diverse domain images through both theoretical analysis and extensive experiments. A natural extension would be its application to learned video compression (LVC). Notably, the intra-frame coding component of LVC models, such as the frame coders in the DCVC series[1-6], is fundamentally a LIC. Therefore, our method could be utilized to directly accelerate the training of these frame coders. However, the motion estimation and temporal modeling components in DCVCs introduce additional complexities that require further investigation. Exploring ways to extend our acceleration approach to these components remains an open challenge for future work."
>
> Thank you once again for your valuable feedback.
>
> Reference:
>
> [1]. Li, Jiahao, Bin Li, and Yan Lu. "Deep contextual video compression." Advances in Neural Information Processing Systems 34 (2021): 18114-18125.
>
> [2]. Sheng, Xihua, et al. "Temporal context mining for learned video compression." IEEE Transactions on Multimedia 25 (2022): 7311-7322.
>
> [3]. Li, Jiahao, Bin Li, and Yan Lu. "Hybrid spatial-temporal entropy modelling for neural video compression." Proceedings of the 30th ACM International Conference on Multimedia. 2022.
>
> [4]. Li, Jiahao, Bin Li, and Yan Lu. "Neural video compression with diverse contexts."
> Proceedings of the IEEE/CVF conference on computer vision and pattern recognition. 2023.
>
> [5]. Li, Jiahao, Bin Li, and Yan Lu. "Neural video compression with feature modulation." Proceedings of the IEEE/CVF Conference on Computer Vision and Pattern Recognition. 2024.
>
> [6]. Jia, Zhaoyang, et al. "Towards Practical Real-Time Neural Video Compression." arXiv preprint arXiv:2502.20762 (2025).

---

> ### Author Response · Authors · 2025-03-24
> **Following up on Rebuttal Submission.**
>
> Dear Reviewer jsnA,
>
> Thank you once again for your valuable feedback. We believe the revisions and clarifications mentioned in our response thoroughly address your concerns regarding the new discussion on video compression and significantly strengthen the evidence supporting our primary claim: "Learned image compression (LIC) model parameters are highly correlated and can be effectively represented by a few distinct 'Modes', which capture their intrinsic dimensions. This allows us to model neural training dynamics in a reduced-dimensional space to accelerate LIC training."
>
> Please don’t hesitate to let us know if there are any remaining concerns or if additional clarification would help improve the manuscript further.
>
> Thank you for your time and consideration.

---

> > ### Comment · Reviewer_jsnA · 2025-04-04
> > **Response**
> >
> > Thank you for the author's response. I consider the revised version of the manuscript to be acceptable.

---

### Review · Reviewer_FNpp · 2025-03-03

**Summary Of Contributions:**

This paper proposes an efficient training framework for Learned Image Compression (LIC) methods by reducing training costs without sacrificing performance. Specifically, the **Sensitivity-aware True and Dummy Embedding Training (STDET)** mechanism clusters parameters and gradually embeds less sensitive ones, lowering the number of trainable parameters. To stabilize this process, a **Sampling-then-Moving Average (SMA)** technique is introduced, ensuring smooth weight updates and reducing training variance.  Overall, the proposed approach accelerates LIC training with fewer parameters and reduced variance, offering insights for scalable and efficient LIC optimization.

**Audience:**

Yes

**Broader Impact Concerns:**

None.

**Claims And Evidence:**

Yes

**Requested Changes:**

Please see weaknesses.

**Strengths And Weaknesses:**

Strengths:
1) The extensive experimental results show that the proposed method is quite effective in reducing training time without sacrificing performance.
2) The theoretical analysis for the proposed STDET is well-studied and promising.


Weaknesses:
1) The technique contribution of SMA is not very clear. While the authors state that "EMA is designed for the testing phase, I do not agree with this statement. A lot of research have applied EMA in this training phase, e.g. [1]. I do not see any technique contribution for the proposed SMA because it is just an EMA on the training phase. I suggest the authors clarify this difference between the proposed SMA and the EMA algorithm.
2) The descriptions in Figure (1) need to be clarified. Why is epoch 70 chosen as the reference stage? Are 70 epochs enough for model convergence? In addition, I recommend adding one more column about epoch 0 to the table in Figure 1 so that the readers can know the parameter change during the whole optimization.
3) Some notations need to be clarified. In Table 3, the meaning of the %value should be clarified in the table, rather than requiring the readers to check it in the main text.


[1] Song, Yang, et al. "Consistency Models." International Conference on Machine Learning. PMLR, 2023.

---

> ### Author Response · Authors · 2025-03-14
> **Response to Reviewer FNpp: Clarification of SMA, updated Fig.1 and improved Table 3 (part 1)**
>
> Dear Reviewer FNpp,
>
> Thank you for recognizing the effectiveness of our proposed method and for acknowledging the thoroughness and promise of our theoretical analysis. Below, we address the points you raised:
>
> **1. Clarification on the difference between the proposed SMA and the EMA algorithm:**
>
> We first sincerely appreciate the reviewer’s reference, which has helped us refine our statements and claims. We revise the initial statement in Section 3.3.2 (Proposed SMA) for improved clarity. For your convenience, we provide the updated version below:
>
> "While EMA is typically used during the testing phase, where its parameters do not influence training and only serve as the final model state, some methods [1,2] integrate EMA into the training process. They achieve this by explicitly computing loss functions that encourage the alignment of SGD trajectories with EMA trajectories, thereby enhancing consistency and stability. In contrast, our approach follows a more straightforward and simpler design. Rather than introducing additional alignment losses, we directly synchronize the EMA-updated sampled SGD parameters (SMA) to the SGD parameters. This enables smoother training dynamics while keeping low complexity."
>
> To further clarify, we provide an illustrative comparison of EMA, explicit alignment methods, and our proposed SMA in Fig. 14 and Appendix Section A.4. The key differences are summarized as follows:
>
> - **EMA:** This method constructs a smooth EMA trajectory based on the SGD updates. EMA weights are typically maintained separately for inference purposes and do not influence SGD weights during training. Consequently, there is no feedback mechanism from EMA weights to SGD weights.
> - **Explicit Alignment Methods [1,2]:** These methods follow the same EMA update rule while introducing an additional loss function to encourage alignment between the EMA and SGD weights. By explicitly minimizing the discrepancy between the outputs of the EMA model and the SGD model, these approaches promote a smoother optimization trajectory. However, they require additional computational overhead due to the extra loss computations and gradient updates.
> - **Proposed SMA:** Our method periodically applies the EMA update rule to a separate SMA weight, derived from sampled SGD parameters. Unlike EMA, which updates in every iteration without influencing the training trajectory, SMA periodically synchronizes its updated weight back to the SGD weight. Our SMA establishes a feedback mechanism that enhances smoothness, while maintaining low computational complexity. Additionally, SMA updates occur at predefined intervals rather than at every iteration, reducing the overhead associated with frequent updates.
>
> These distinctions highlight that while both explicit alignment and SMA integrate EMA into training to enhance stability, SMA achieves this in a more computationally efficient and straightforward manner. Instead of relying on explicit loss terms and gradient updates, SMA interpolates sampled SGD parameters and synchronizes SMA weight with the SGD weight, offering a simple yet effective approach to smoothing the training trajectories.
>
> **2. Why is epoch 70 used in Figure 1? Additionally, please add a column for epoch 0:**
>
> We sincerely appreciate the reviewer’s insightful comment. Upon further consideration, we recognize that using epoch 70 was not an ideal choice, as the model had not yet fully converged at that point. To provide a more representative comparison, we have now updated it to reflect epoch 120, which corresponds to the converged state.
>
> Additionally, we have incorporated a new column for epoch 0. As expected, the majority of the coefficients initialized at epoch 0 exhibit significant changes by epoch 120. In contrast, for other epochs, the trend remains consistent with the previous epoch 70 setting—most coefficients either remain stable or undergo only minor adjustments as training progresses from epoch 20 to 120, with slight numerical variations.
>
> Notably, the proportion of coefficients in the 0\% to 1\% interval increases significantly, rising from 31.64\% at epoch 20 to 81.70\% at epoch 100. This indicates a marked stabilization of the affine coefficients over time.
>
> For your convenience, we put the updated table here:
>
> | Change | 0 | 20 | 40 | 60 | 80 | 100 |
> |--------|---|----|----|----|----|-----|
> | 0% ~ 1% | 3.85% | 31.64% | 41.61% | 61.70% | 72.84% | 81.70% |
> | 1% ~ 2% | 0.21% | 9.22% | 8.01% | 7.38% | 6.00% | 5.81% |
> | 2% ~ 5% | 0.62% | 19.04% | 18.35% | 10.92% | 5.24% | 4.02% |
> | 5% ~ 10% | 1.04% | 15.78% | 12.87% | 9.60% | 7.69% | 4.11% |
> | 10% ~ 20% | 2.12% | 11.10% | 9.90% | 5.21% | 4.77% | 2.44% |
> | 20% ~ 50% | 7.29% | 10.89% | 7.99% | 4.33% | 2.93% | 1.65% |
> | 50% ~ 100% | 84.87% | 2.33% | 1.27% | 0.86% | 0.53% | 0.27% |

---

> ### Author Response · Authors · 2025-03-14
> **Response to Reviewer FNpp: Clarification of SMA, updated Fig.1 and improved Table 3 (part 2)**
>
> **3. Notation In Table 3:**
>
> We thank the reviewer for pointing out this. The \% represent BD-Rate. We update Table 3, 4, 5 by explicitly stating the meaning of the \%
>
> Thank you once again for your valuable feedback.
>
> Reference:
>
> [1]. He, Kaiming, et al. "Momentum contrast for unsupervised visual representation learning." Proceedings of the IEEE/CVF conference on computer vision and pattern recognition. 2020.
>
> [2]. Song, Yang, et al. "Consistency models." (2023).

---

> ### Author Response · Authors · 2025-03-24
> **Following up on Rebuttal Submission.**
>
> Dear Reviewer FNpp,
>
> Thank you once again for your valuable feedback. We believe the revisions and clarifications mentioned in our response thoroughly address your concerns regarding the clarification of SMA, Fig.1 and Table 3, and significantly strengthen the evidence supporting our primary claim: "Learned image compression (LIC) model parameters are highly correlated and can be effectively represented by a few distinct 'Modes', which capture their intrinsic dimensions. This allows us to model neural training dynamics in a reduced-dimensional space to accelerate LIC training."
>
> Please don’t hesitate to let us know if there are any remaining concerns or if additional clarification would help improve the manuscript further.
>
> Thank you for your time and consideration.

---

### Review · Reviewer_EKLo · 2025-03-05

**Summary Of Contributions:**

This work accelerates learned image compression (LIC) training by modeling neural training dynamics. The authors propose **STDET**, which clusters parameters into modes and progressively embeds non-reference parameters based on stability, reducing trainable parameters. They further introduce SMA, a moving average technique to smooth training and minimize variance. Theoretical analysis and experiments show significant training speedup without performance loss.

**Audience:**

Yes

**Claims And Evidence:**

Yes

**Requested Changes:**

- The proposed STDET mechanism clusters parameters into modes and embeds non-reference parameters dynamically, but its effectiveness across different LIC architectures remains unclear. The authors should provide additional empirical evidence on a broader set of models to demonstrate generalizability.

- The paper would benefit from more comprehensive ablation studies to analyze the impact of key components such as the embedding threshold, number of modes, and sensitivity estimation strategy. This would strengthen claims regarding the effectiveness of STDET and SMA.

- While comparisons with standard SGD and a few subspace methods are provided, additional baselines such as sparsity-based training techniques or recent efficient optimization methods for deep learning could provide further insight into the advantages and limitations of the proposed approach.

- Since the method relies on progressively reducing the number of trainable parameters, an analysis of potential instability or performance degradation in later training stages would be useful. Providing learning curves with variance analysis could help assess this.

- The paper highlights reduced training time, but a breakdown of computational savings in terms of FLOPs, memory usage, or energy consumption would add more depth to the efficiency discussion.

**Strengths And Weaknesses:**

**Strengths:**
(+) This work addresses a crucial challenge in accelerating the training of learned image compression (LIC) models, which are typically computationally expensive.

(+) The proposed STDET mechanism effectively reduces the number of trainable parameters by leveraging parameter correlations, leading to improved efficiency.

(+) The SMA technique provides a well-motivated strategy to stabilize training and minimize variance, ensuring smoother optimization.

(+) Theoretical analysis and extensive experiments on multiple LIC models demonstrate the effectiveness of the approach in reducing training time without sacrificing performance.

**Weaknesses:**
(-) The extent to which the proposed STDET mechanism generalizes across different LIC architectures remains unclear.

(-) While the method reduces training costs, the core idea of parameter clustering and embedding follows existing low-dimensional training strategies, limiting technical novelty.

(-) The ablation studies could be expanded to provide a more detailed analysis of the impact of different hyperparameter choices on the effectiveness of STDET and SMA.

---

> ### Author Response · Authors · 2025-03-14
> **Response to Reviewer EKLo: Added RAW image compression, additional sparsity-based training techniques, clarification for hyperparameters selection, new ablation study and FLOPs discussion. (part 1))**
>
> Dear Reviewer EKLo,
>
> Thank you for your valuable feedback and for recognizing the practical significance of our work. Below, we address the points raised:
>
> **1. Additional empirical evidence on a broader set of models to demonstrate generalizability:**
>
> The authors thank the reviewer for the comments.
> We have selected eight LIC models that span five distinct image domains and encompass recent representative open-source LIC methods. To the best of our knowledge, no prior work including [1]–[8] has conducted such a comprehensive evaluation. This extensive validation provides strong empirical evidence of our method’s generalizability. More specifically, we conducted experiments on ELIC[1], TCM[2], and FLIC[3] for natural image compression, which is the primary focus and scope of this paper in the previous version. Additionally, our method is evaluated on ECSIC[4] and BiSIC[5] for stereo image compression, HL-RSCompNet[6] for remote sensing image compression, and SFTIP[7] for screen content image compression to demonstrate its effectiveness and generalizability.
>
> To further address the reviewer's concern, we have now incorporated additional experiments on R2LCM[8] for RAW image compression. The new results, presented in Section A.1.4 (RAW Image Compression), exhibit the same trend as our previous findings, confirming that our method consistently accelerates the training process across different models.
>
> If the reviewer is interested in specific LIC results beyond those presented, we would be more than happy to discuss and provide further insights.

---

> ### Author Response · Authors · 2025-03-14
> **Response to Reviewer EKLo: Added RAW image compression, additional sparsity-based training techniques, clarification for hyperparameters selection, new ablation study and FLOPs discussion. (part 2))**
>
> **2. More comprehensive ablation studies:**
>
> We sincerely thank the reviewer for the comment. A thorough ablation study is presented in Section 4.4 (Ablation study), covering all the hyperparameters of the proposed method. Many of the ablation studies requested by reviewers were already included in the initial submission. However, to further address your concerns, we provide the following clarifications and additional experiments:
>
> **(1) Embedding Threshold**
>
> The embedding threshold is determined by the embedding percentage $ P $ and period $ L $. The results are shown in 4.4.2 (What is the impact of factors of the embedding mechanism?). In the original submission, we evaluated the following settings:
>
> - $ P = 1$\%, $ L = 1 $, corresponding to a total threshold of 50% (final setting).
> - $ P = 1$\%, $ L = 2 $, corresponding to a total threshold of 25%.
> - $ P = 2 $\%, $ L = 1 $, corresponding to a total threshold of 75%, as mentioned in the paper, where only 75% of the parameters are deemed embeddable, focusing on those that are less sensitive. 75% is reached at the intermediate training stage.
>
> In addition to these, we now provide an additional experiment:
>
> - $ P = 1.5 $\%, $ L = 1 $, which also results in a total threshold of 75% but differs from $ P = 2 $\%, $ L = 1 $ in that the 75% threshold is reached only at the final epoch.
>
> The results are presented in Tables 3, 4, and 5 in Section 4.4 (Ablation study). For the reviewer’s convenience, we summarize them here:
>
> | Embedding Setting     | ELIC   | TCM-S  | FLIC   | Description                   |
> | --------------------- | ------ | ------ | ------ | ----------------------------- |
> | $P = 1$% & $L = 1$   | 0%     | 0%     | 0%     | final setting (50% threshold) |
> | $P = 1$% & $L = 2$   | -1.40% | -0.83% | -0.92% | 25% threshold                 |
> | $P = 1.5$% & $L = 1$ | 5.20%  | 5.93%  | 5.48%  | 75% threshold (final epoch)   |
> | $P = 2$% & $L = 1$   | 10.72% | 10.42% | 10.36% | 75% threshold (intermediate)  |
>
> As observed, our final setting of $ P = 1$%, $ L = 1 $ achieves the best balance between final performance and the embedded parameters percentage.
>
> **(2) Number of Modes**
>
> This ablation study was already included in the initial submission, where we performed a grid search over this parameter and found an efficient way to determine the best hyperparameters in real application. The details can be found in Section 4.4.4 (How to Select the Number of Modes and Sampled Parameters). We kindly refer the reviewer to that section for more information.
>
> **(3) Sensitivity Estimation Strategy**
>
> Our method employs a hybrid approach that combines accurate layer-wise assessment with rough parameter-wise estimation. To further demonstrate its effectiveness, we present two additional comparisons:
>
> - **Accurate parameter-wise assessment:** This method provides a highly precise evaluation; however, it is impractical for real-world applications due to the immense computational cost associated with ranking and permuting millions of parameters. We include this result to illustrate the effectiveness of our approach.
> - **Rough first order parameter-wise estimation:** This is the widely adopted approach in previous research. Comparison to this baseline highlights the advantages of our proposed method.
>
> The results for these comparisons are provided in Section 4.4 (Ablation study). For ease of reference, we summarize them here:
>
> | Strategy        | ELIC   | TCM-S  | FLIC   | Description                              |
> | --------------- | ------ | ------ | ------ | ---------------------------------------- |
> | Combined method | 0%     | 0%     | 0%     | Hybrid approach (final proposed setting) |
> | w/ Accurate     | -1.21% | -1.38% | -1.35% | Accurate parameter-wise assessment       |
> | w/ First order  | 4.55%  | 5.16%  | 5.11%  | Rough first order parameter-wise est.    |
> | w/o Sensitivity | 12.34% | 10.42% | 10.73% | No sensitivity estimation                |
>
> The results indicate that while accurate parameter-wise assessment offers only slight improvements, it comes at an impractically high computational cost. Conversely, rough first-order parameter-wise estimation significantly degrades performance. This analysis underscores the effectiveness of our proposed combined approach, which strikes a balance between computational efficiency and model performance. Additionally, the absence of sensitivity estimation reveals that certain parameters are highly sensitive, and improper embedding leads to severe performance degradation, further validating the effectiveness of our sensitivity estimation method.
>
> These ablation studies collectively provide a comprehensive validation of the effectiveness of our proposed method

---

> ### Author Response · Authors · 2025-03-14
> **Response to Reviewer EKLo: Added RAW image compression, additional sparsity-based training techniques, clarification for hyperparameters selection, new ablation study and FLOPs discussion. (part 3))**
>
> **3. Additional sparsity-based training techniques:**
>
> We greatly appreciate the reviewer’s thoughtful comments. We first want to emphasize that our method is fundamentally different from sparsity-based training approaches [9,10]. These methods primarily aim to train sparse models for efficient inference; however, their training speed is the same or even slower than standard training methods due to platform constraints and additional steps required for maintaining sparsity.
>
> To provide a holistic comparison of the advantages of our proposed method, we have included two recent representative sparsity-based training methods: RigL and SRigL. As demonstrated in Table 2 (Comparison with Various Efficient Training Methods) in the main content, these methods also fail to achieve satisfactory results.
>
> The results are also presented in the Table below:
>
> | Settings        | Training epoch ↓ | Training space dimension ↓ | Total training time ↓ | Inference FLOPs ↓ | R-D loss ↓ |
> | --------------- | ---------------- | -------------------------- | --------------------- | ----------------- | ---------- |
> | ELIC + SGD      | 120              | 35,424,505                 | 38h                   | 437.6G            | 0.3444     |
> | ELIC + P-SGD    | **70**           | **40**                     | **23h**               | 437.6G            | div.       |
> | ELIC + P-BFGS   | **70**           | **40**                     | 35h                   | 437.6G            | 0.3982     |
> | ELIC + TWA      | **70**           | 50                         | **23h**               | 437.6G            | div.       |
> | ELIC + Rig      | **70**           | 35,424,505                 | **23h**               | 218.8G            | 0.3624     |
> | ELIC + SRigL    | **70**           | 35,424,505                 | 24h                   | 214.4G            | 0.3601     |
> | ELIC + Proposed | **70**           | 50*                        | **23h**               | 437.6G            | **0.3442** |
>
> **Train Conditions**: 1 × Nvidia 4090 GPU, i9-14900K CPU, 128GB RAM. "div." indicates that these methods result in loss divergence, eventually causing the training to crash. *: The training dimension of our proposed method continues to decrease as training proceeds, theoretically converging to 50. λ = 0.0018. R-D loss = λ · 255² · MSE + Bpp. **Bold** indicates the best.
> FLOPs are calculated using the ptflops library with an input image size of 512×512 pixels.
>
> We can see that under the same number of training epochs, their R-D loss is significantly higher than both our method and standard training. Furthermore, due to the pruning and regrowth mechanism inherent in these methods, the training dimensionality is not reduced. While they do offer the advantage of lowering inference-time FLOPs, this benefit is rendered ineffective given their poor performance.
>
>
> **4. Variance analysis:**
>
> We appreciate the reviewer’s review. In the initial submission, we already provided a variance analysis both empirically (Section A.2: Variance Reduction) and theoretically (Section A.6: Noisy Quadratic Analysis). Both the empirical results and theoretical analysis have demonstrated that our method exhibits lower variance compared to standard training.
>
> To further address the reviewer's concern, we have added an additional figure in Section A.2 (Variance Reduction) illustrating the R-D loss trend of ELIC models along with its standard deviation range. The results clearly demonstrate that our method shows lower standard deviation (also for variance) than standard training. We kindly refer the reviewer to this section for further details.
>
>
>
> **5. Efficiency:**
>
> We appreciate the reviewer's suggestion. In response, we have now included inference FLOPs in Table 2 (Comparison with Various Efficient Training Methods). Our analysis confirms that neither our method nor other low-dimensional training methods impact FLOPs, whereas sparsity-based training methods, while not accelerating training, do reduce inference FLOPs.
>
> Additionally, memory consumption has already been discussed in Section A.5 (Limitations and Future Work). Our method requires only a negligible amount of additional memory compared to standard training.
>
> Regarding energy consumption, there is currently no universally recognized method for its measurement. Therefore, we have not included additional discussions on this aspect. However, if the reviewer has a specific methodology in mind, we would be happy to conduct the analysis accordingly.
>
> Thank you once again for your valuable feedback.

---

> ### Author Response · Authors · 2025-03-14
> **Response to Reviewer EKLo: Added RAW image compression, additional sparsity-based training techniques, clarification for hyperparameters selection, new ablation study and FLOPs discussion. (part 4))**
>
> **References**
>
> [1]. He, Dailan, et al. "Elic: Efficient learned image compression with unevenly grouped space-channel contextual adaptive coding." Proceedings of the IEEE/CVF Conference on Computer Vision and Pattern Recognition. 2022.
>
> [2]. Liu, Jinming, Heming Sun, and Jiro Katto. "Learned image compression with mixed transformer-cnn architectures." Proceedings of the IEEE/CVF conference on computer vision and pattern recognition. 2023.
>
> [3]. Li, Han, et al. "Frequency-Aware Transformer for Learned Image Compression." The Twelfth International Conference on Learning Representations.
>
> [4]. Wödlinger, Matthias, et al. "ECSIC: Epipolar cross attention for stereo image compression." Proceedings of the IEEE/CVF Winter Conference on Applications of Computer Vision. 2024.
>
> [5]. Liu, Zhening, et al. "Bidirectional stereo image compression with cross-dimensional entropy model." European Conference on Computer Vision. Cham: Springer Nature Switzerland, 2024.
>
> [6]. Xiang, Shao, and Qiaokang Liang. "Remote sensing image compression based on high-frequency and low-frequency components." IEEE Transactions on Geoscience and Remote Sensing 62 (2024): 1-15.
>
> [7]. Zhou, Fangtao, et al. "Enhanced Screen Content Image Compression: A Synergistic Approach for Structural Fidelity and Text Integrity Preservation." Proceedings of the 32nd ACM International Conference on Multimedia. 2024.
>
> [8]. Wang, Yufei, et al. "Beyond learned metadata-based raw image reconstruction." International Journal of Computer Vision 132.12 (2024): 5514-5533.
>
> [9]. Evci, Utku, et al. "Rigging the lottery: Making all tickets winners." International conference on machine learning. PMLR, 2020.
>
> [10]. Lasby, Mike, et al. "Dynamic Sparse Training with Structured Sparsity." The Twelfth International Conference on Learning Representations.

---

> ### Author Response · Authors · 2025-03-24
> **Following up on Rebuttal Submission.**
>
> Dear Reviewer EKLo,
>
> Thank you once again for your valuable feedback. We believe the revisions and clarifications mentioned in our response thoroughly address your concerns regarding the broader set of models, additional sparsity-based training techniques, clarification for hyperparameter selection, new ablation study, and FLOPs discussion, and significantly strengthen the evidence supporting our primary claim: "Learned image compression (LIC) model parameters are highly correlated and can be effectively represented by a few distinct 'Modes', which capture their intrinsic dimensions. This allows us to model neural training dynamics in a reduced-dimensional space to accelerate LIC training."
>
> Please don’t hesitate to let us know if there are any remaining concerns or if additional clarification would help improve the manuscript further.
>
> Thank you for your time and consideration.

---

### Review · Reviewer_Qp3R · 2025-03-07

**Summary Of Contributions:**

The authors propose Sensitivity-aware True and Dummy Embedding Training mechanism (STDET) for Learned Image Compression (LIC) which is combined with Sampling-then-Moving Average (SMA) to regularize training state variances.
They separate parameters into nodes and then successively embeds lower sensitivity parameters.
By adaptively reducing the number of parameters during training, they achieve faster convergence without sacrificing performance.

**Audience:**

Yes

**Claims And Evidence:**

Yes

**Requested Changes:**

- Regarding the hyperparamers, is there a (structured) way to find good hyperparameters for new data distributions, i.e. is there a better approach than brute force (grid search).? If the authors have intuition about this it would be good to add.

**Strengths And Weaknesses:**

Strength:
- The proposed method provides significant speedup compared to other models in the literature at without sacrificing performance
- They provide a theoretical foundation analyzing the noisy quadratic model
- They provide ample amount of experiments to validate their claims

Weaknesses:
- It is not clear how much of an impact the choice of hyperparameters has on performance, so it is not obvious whether the proposed method would perform as well on other practical data
- Regarding technical novelty, the key ideas behind their method are well known in the literature, so on the purely technical side the contribution is limited

---

> ### Author Response · Authors · 2025-03-14
> **Response to Reviewer Qp3R: Clarification for ablation study and hyperparameters selection.**
>
> Dear Reviewer Qp3R,
>
> Thank you very much for your insightful feedback and positive assessment of our proposed method. We appreciate your recognition of the significant training speedup achieved by our method and the robustness of our experimental validation. Below we address each of your specific comments in detail:
>
> **1. Impact of hyperparameter choice:**
>
> The authors thank the reviewer for the comments. We have conducted a comprehensive analysis on the impact of hyperparameter selection in Section A.1 (Hyperparameter selection) in the initial submission, which is renamed to Section 4.4 Ablation study to avoid confusion. Specifically, Section 4.4.4 (How to select the number of modes, and the number of sampled parameters?) describes a structured and efficient approach for selecting the appropriate number of modes and sampled trajectories. This approach avoids exhaustive grid search and instead relies on evaluating the instant cmd model performance through few forward passes. Additionally, we extensively demonstrate our method's effectiveness across various practical image compression domains including natural images (Section 4), stereo images, remote sensing images, screen content images, and newly added RAW images (Section A.1). These results strongly validate the generalizability and robustness of our hyperparameter selection strategy.
>
> **2. Technical novelty (well-known ideas):**
>
> The authors appreciate the reviewer’s insightful observations. We agree that the underlying concept of low-dimensional hypothesis is well-known. However, designing a method for LIC based on this concept is nontrivial and inherently challenging, given LIC's significantly higher complexity compared to tasks like image classification (e.g., CIFAR-10/100). As highlighted in Table 2 (Comparison with Various Efficient Training Methods), direct application of existing efficient training methods developed for simpler tasks does not yield satisfactory performance in LICs, e.g., P-SGD and TWA leads to loss divergence, and  P-BFGS, RigL, and SRigL leads to significant R-D loss increase without accelerating training. Our method bridges this gap, making low-dimensional training practically beneficial for LIC by achieving substantial training speedups while maintaining competitive or even superior compression performance. We believe this advancement provides meaningful technical value to the LIC community.
>
> **3. Structured approach to hyperparameter selection for new data distributions:**
>
> The authors are grateful for the reviewer’s feedback.
> Indeed, we have a structured and efficient methodology for hyperparameter selection, explicitly detailed in Section 4.4.4. A key observation is that the number of modes and sampled trajectories must be carefully chosen for each model, with larger models generally requiring more modes and sampled trajectories.
>
> Building on this insight, our hyperparameter selection method leverages a strong correlation between instant CMD performance and final model accuracy, effectively eliminating the need for impractical brute-force grid searches during training. By requiring only a few forward evaluations, our approach enables efficient and practical hyperparameter selection. The effectiveness of this method has been extensively validated in both Section 4 (natural images) and Section A.1 (other practical image domains), consistently demonstrating its efficiency, robustness, and broad applicability.
>
>
>
> Thank you once again for your valuable feedback.

---

> ### Author Response · Authors · 2025-03-24
> **Following up on Rebuttal Submission.**
>
> Dear Reviewer Qp3R,
>
> Thank you once again for your valuable feedback. We believe the revisions and clarifications mentioned in our response thoroughly address your concerns regarding the ablation study and hyperparameter selection and significantly strengthen the evidence supporting our primary claim: "Learned image compression (LIC) model parameters are highly correlated and can be effectively represented by a few distinct 'Modes', which capture their intrinsic dimensions. This allows us to model neural training dynamics in a reduced-dimensional space to accelerate LIC training."
>
> Please don’t hesitate to let us know if there are any remaining concerns or if additional clarification would help improve the manuscript further.
>
> Thank you for your time and consideration.

---

### Decision · Action_Editor_cAjU · 2025-04-12

**Recommendation:** Accept as is

**Comment:**

The reviewers agree that the paper brings useful insights to the LIC domain by accelerating training without compromising performance. While the core techniques (e.g., parameter embedding, smoothing) build on existing ideas, their combination and adaptation are novel in this context. Theoretical reasoning is sound, and experiments show consistent efficiency gains. Some concerns were raised around generalizability and marginal performance improvements, but these don't outweigh the practical value of the contribution.

**Audience:**

Researchers working on learned image compression and efficient training methods would likely find the proposed approach interesting, especially given the practical speedup without loss in performance.

**Claims And Evidence:**

The paper provides solid evidence that the proposed method can significantly speed up training in the LIC setting, with clear experiments and theoretical insights to back it up. The efficiency gains are well demonstrated across multiple setups. Some parts are less clear—especially how much hyperparameter tuning is needed if applied to new datasets or tasks. Also, while training is faster, the final performance gains (e.g., in BD-Rate) are quite small. So overall, the claims around efficiency are well supported, but broader generalization or performance improvements would benefit from more evidence.